# Non-Muscle Myosin II A: Friend or Foe in Cancer?

**DOI:** 10.3390/ijms25179435

**Published:** 2024-08-30

**Authors:** Wasim Feroz, Briley SoYoung Park, Meghna Siripurapu, Nicole Ntim, Mary Kate Kilroy, Arwah Mohammad Ali Sheikh, Rosalin Mishra, Joan T. Garrett

**Affiliations:** 1Department of Pharmaceutical Sciences, James L. Winkle College of Pharmacy, Cincinnati, OH 45229, USA; ferozwm@mail.uc.edu (W.F.); sarahpark0806@gmail.com (B.S.P.); meghna.siripurapu24@ihsd.us (M.S.); ntimne@mail.uc.edu (N.N.); kilroymk@mail.uc.edu (M.K.K.); rosalin.mishra@cchmc.org (R.M.); 2Cancer Research Scholars Program, College of Allied Health Sciences, University of Cincinnati, Cincinnati, OH 45267, USA; 3Division of Endocrinology, University of Cincinnati, Cincinnati, OH 45267, USA; sheikhaw@ucmail.uc.edu

**Keywords:** myosin IIA, non-muscle myosin IIA, *MYH9*, motor protein, tumorigenesis

## Abstract

Non-muscle myosin IIA (NM IIA) is a motor protein that belongs to the myosin II family. The myosin heavy chain 9 (*MYH9*) gene encodes the heavy chain of NM IIA. NM IIA is a hexamer and contains three pairs of peptides, which include the dimer of heavy chains, essential light chains, and regulatory light chains. NM IIA is a part of the actomyosin complex that generates mechanical force and tension to carry out essential cellular functions, including adhesion, cytokinesis, migration, and the maintenance of cell shape and polarity. These functions are regulated via light and heavy chain phosphorylation at different amino acid residues. Apart from physiological functions, NM IIA is also linked to the development of cancer and genetic and neurological disorders. *MYH9* gene mutations result in the development of several autosomal dominant disorders, such as May-Hegglin anomaly (MHA) and Epstein syndrome (EPS). Multiple studies have reported NM IIA as a tumor suppressor in melanoma and head and neck squamous cell carcinoma; however, studies also indicate that NM IIA is a critical player in promoting tumorigenesis, chemoradiotherapy resistance, and stemness. The ROCK-NM IIA pathway regulates cellular movement and shape via the control of cytoskeletal dynamics. In addition, the ROCK-NM IIA pathway is dysregulated in various solid tumors and leukemia. Currently, there are very few compounds targeting NM IIA, and most of these compounds are still being studied in preclinical models. This review provides comprehensive evidence highlighting the dual role of NM IIA in multiple cancer types and summarizes the signaling networks involved in tumorigenesis. Furthermore, we also discuss the role of NM IIA as a potential therapeutic target with a focus on the ROCK-NM IIA pathway.

## 1. Introduction

Cytoskeletal proteins, such as actin filaments (7–8 nm), intermediate filaments (10 nm), and microtubules (24 nm), are important for cell motility, contraction, polarity, the maintenance of cell morphology, and mechanical support during cytokinesis [1]. Actin filaments and microtubules maintain the directionality of intracellular transport and serve as paths for motor proteins, including myosin, dyneins, and kinesins [1,2]. Intermediate filaments absorb mechanical stress and provide integrity for the cytoskeleton [3]. Myosin motor proteins are molecular machines that perform various motor functions, which require mechanical force and movement. There are more than 30 unique classes of proteins in the myosin superfamily with different distribution patterns and functions [4,5,6]. Non-muscle myosin IIA (NM IIA) is a hexamer composed of three pairs of peptides: a dimer of heavy chains (230 kDa), essential light chains (ELCs, 17 kDa), and regulatory light chains (RLCs, 20 kDa).

In smooth and striated muscles, myosin regulates muscle contraction, whereas non-muscle myosin (NMII) regulates cell migration, adhesion, cytokinesis, and polarity. NMIIs are also important for the growth and differentiation of muscle cells [7]. Additionally, they have a distinct function in the maintenance of smooth muscle tension [8,9]. Myosin binds to actin filaments and induces the magnesium-dependent hydrolysis of ATP to generate mechanical force and tension that propels the sliding of actin filaments. This mechanical force is used for cell motility and division, the intracellular transport of molecules, the maintenance of cell polarity, and signal transduction [10,11]. The cellular functions of myosin proteins vary greatly due to differences in their enzymatic activity and regulation. A total of 38 genes encode various myosin proteins and are divided into 12 classes based on their sequence similarities [11].

Vertebrates have three different isoforms of non-muscle myosin II heavy chains: NMHC IIA, NMHC IIB, and NMHC IIC [12]. The genomes of *Saccharomyces cerevisiae* (yeast), *Drosophila melanogaster* (fruit fly), and *Dictyostelium discoideum* (amoeba) carry a single myosin heavy chain (*MHC*) gene encoding the NMHC II isoform [13,14,15]. A structural analysis revealed 60–80% identity in the primary sequence of NMHC IIs with the highest similarity in the head domains [16,17]. NMHC IIs combined with myosin light chains (ELC plus RLC) are denoted as NM IIA, NM IIB, and NM IIC, respectively. Although NMII isoforms share sequence and structural similarity, they differ significantly in their motor head ATPase activity, the rate of assembly of myosin filaments, and in the time spent attaching to actin filaments [18,19,20]. Furthermore, each isoform exhibits different patterns of expression in tissues with distinct intracellular distributions [21,22,23,24,25,26]. However, it is important to note that they also share overlapping functions; for example, all NMII isoforms efficiently contribute to cytokinesis [27].

NM IIA is involved in embryogenesis, organogenesis, and immunity; however, its critical role in cancer, autosomal dominant disorders, cataracts, infections, kidney and vascular diseases, and neuronal disorders is gaining importance. This study provides an in-depth review of the structure, function, and regulation of NM IIA. We focus on understanding the role and significance of NM IIA in tumorigenesis as some studies have reported it to be a tumor suppressor, whereas others have shown it to be a tumor promoter. Furthermore, we address the pathological consequences of *MYH9* mutations. Figure 1 shows the frequency of *MYH9* mRNA expression and alterations across 32 pancancer studies obtained from cBioPortal (www.cbioportal.org). Finally, we address the potential role of NM IIA as a therapeutic target, with a particular focus on targeting the ROCK-NM IIA pathway.

## 2. Structure of NM IIA

The human *MYH9* (myosin heavy chain 9) gene is located on chromosome 22q12-13 and is composed of 41 exons. It is a widely expressed housekeeping gene with a high GC content and no TATA boxes. *MYH9* in mice encodes a protein that is 97% identical to human NMHC IIA [28]. For this review, we used the following nomenclature: *MYH9*—a gene; NMHC IIA—a protein produced by the *MYH9* gene; and NM IIA—a fully functional hexamer composed of two homodimers of NMHC IIA heavy chains, two ECLs (*MYL6* gene), and two RLCs (*MYL 12A/B* genes) that stabilize the heavy chain structure [28] (Figure 2). All myosin II complexes, including the isoforms, have both ELC and RLC, which are essential for its function and regulation. Myosin complex assembly involves complicated machinery. For example, in the Golgi apparatus, UNC-45/CRO1/She4 (UCS) chaperones regulate the folding of heavy chain peptides of the complex. Furthermore, the UCS chaperones are evolutionarily conserved and promote myosin-related functions including cytokinesis, muscle development, endocytosis, and RNA transport. In *Caenorhabditis elegans*, UCS chaperones regulate myosin II complex function to ensure proper filament assembly [29].

The heavy chain of NM IIA is divided into two regions: the N and C termini. The N terminus comprises a globular motor head domain, the functional engine of myosin II, which can bind to actin and generate force via MgATPase activity (with Mg^2+^ serving as an essential cofactor) [30,31]. The neck region, which follows the head region, binds to both myosin light chains (ELC and RLC) [32]. The tail domain or the C terminus is important for homodimerization with other myosin in a helical fashion [33]. Despite showing significant sequence similarity, these regions exhibit different binding affinities for actin filaments. Due to this differential actin binding, different isoforms carry out mechanical work in the cell with varying energy efficiencies [34]. The motor head is composed of four subdomains: the N terminal SH3-like motif, upper subdomain, lower subdomain, and the converter region [34].
Figure 2This figure illustrates the two assembly states of non-muscle myosin II (NMII): the 10S assembly-incompetent state and the 6S assembly-competent state. In the 10S assembly-incompetent state (**left**), the myosin molecule is folded, with the globular head and heavy chain regions interacting through various tail-binding sites, leading to a compact structure. Many intramolecular interactions keep the 10S state in an inactive stable form (Table 1). The interactions involve the Blocked Head (BH), which is the myosin head prevented from binding to actin, and the Free Head (FH), another myosin head that is inhibited but not directly involved in actin binding in the 10S state. The transition to the 6S assembly-competent state (**right**) occurs upon the phosphorylation of the regulatory light chain (RLC), resulting in an extended, active conformation where the heavy chain regions are aligned, allowing for actin binding and ATPase activity, which are essential for NMII’s role in cell contractility and motility. ELC: Essential Light Chain; RLC: Regulatory Light Chain; FH: Free Head; BH: Blocked Head; TF: Tail–Free Head Interaction; TB: Tail–Blocked Head Interaction; TT: Tail–Tail Interaction.
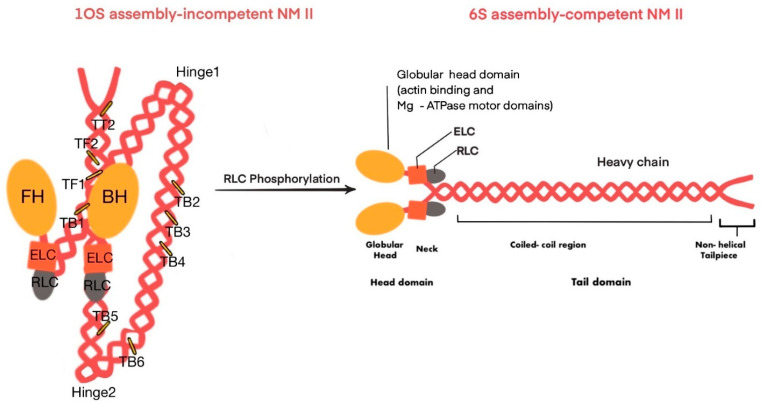


The neck region is like a lever arm that transforms the force produced by the rotation of the motor head into mechanical work. Furthermore, an extended helix with two conserved IQ motifs (IQxxxRGxxxR) is present in the neck region. However, different classes of myosin possess different numbers of IQ motifs [35]. IQ motifs form a hydrophobic seven-turn α-helix chain that can bind to light chains (RLC or ELC) in a Ca^2+^-independent manner. ELC binds the first IQ motif that is located closer to the N-terminus, providing stability to the non-muscle heavy chain (NMHC). RLC binds to the adjacent IQ-like motif that is closer to the coiled coil region, providing both stability and functional regulation to NMHC. The phosphorylation of serine 19 (Ser 19) on the RLC increases the motor head ATPase activity and contributes to filament assembly [16]. The length of the neck region directly impacts myosin II motor function [36].

Various genes encode myosin light chains (MLCs) in humans. For example, *MYL9*, *MYL12A*, and *MYL12B* encode RLCs, whereas *MYL6* and *MYL6B* encode ELCs [37]. Alternative splicing is crucial for regulating gene expression and enhancing protein diversity in eukaryotes. Both *MYL9* and *MYL6* undergo alternative splicing to diversify NMII multimeric complexes. Studies have shown that ELC *MYL6* might interact specifically with NMHC IIC to regulate it both spatially and temporally. Thus, a specific function is assigned to NMHC IIC in addition to its traditional mechanical and kinetic functions [18,38]. The coiled coil region of C-terminus homodimerizes to form a single rod-like structure. These rod-like structures can interact with other myosin proteins to assemble as bipolar filaments, which re nucleation sites for the formation of larger filaments measuring approximately 300 nm in length [39,40]

The proline kink in the helix located at the junction of the head and rod domains allows the myosin II complex to fold into a compact shape. When the RLC is unphosphorylated or inactive, the head and tail domains interact to create a compact conformation of myosin II complex [41,42]. The inactive 10S compact structure of myosin II sediments at 10 S (Svedberg) and does not possess any actin-activated ATPase activity [43,44]. The serine 19 (Ser19) phosphorylation of RLC by the calcium–calmodulin–myosin light chain kinase (MLCK) pathway [45] results in the unfolding of the 10S structure and subsequent formation of the assembly-competent 6S. The 6S form sediments at 6 Svedberg. In their active state, the tail domain assembles in an anti-parallel fashion into ~300 nm long bipolar thick filaments. These bipolar filaments crosslink actin filaments to create different meshes of actomyosin bundles. In the absence of RLC, the tail domains form disorganized filaments. Therefore, the RLC is important to maintain a non-aggregated myosin state that can form filaments and does not directly contribute to ATP hydrolysis [46].

All of the paralogs of NMII can hydrolyze ATP and participate in different cellular processes; however, the rate of hydrolysis differs. For example, compared with muscle myosin II, NM IIA hydrolyzes ATP at a slower rate [47]. Furthermore, the binding time to actin filaments is also low, which might be the reason for the low stability of NM IIA mini-filaments [48]. Vincente et al. reported the continuous assembly and disassembly of NM IIA filaments during membrane protrusion [49]. In contrast, NM IIB and NM IIC exhibit fast kinetics of ATPase hydrolysis, which makes them ideal for force generation in non-muscle cells [50,51]. Therefore, the turnover and assembly rates of NM IIB and NMIIC are slower than those of NM IIA [52].

## 3. Regulation of NM IIA

The specific group of proteins that regulate myosin activity in skeletal and cardiac muscles includes the troponin–tropomyosin complex, myosin-binding protein C (MyBP-C in cardiac muscle), myosin-binding protein H (MyBP-H in skeletal muscle), myosin light chain kinase (MLCK), protein kinase A (PKA) and protein kinase C (PKC), and calcium ions (Ca^2+^) [53] (Figure 3). The three paralogs, NM IIA, NMIIB, and NM IIC, display 60–80% of amino acid sequence similarity and share a common quaternary structure. However, in a homeostatic state, these three paralogs differ in their regulatory mechanisms [54].

### 3.1. Role of RLC Phosphorylation in Regulating NM IIA Activity

As discussed earlier, NM IIA can assume two different conformations: the 10S form in which NM IIA is compact and inactive, and the 6S form, which is the unfolded active form. In the 10S inactive folded form, the important biological properties of NMII, including ATPase hydrolysis activity, bipolar filament formation, and actin filament sliding, are inhibited [55,56,57]. RLCs undergo reversible phosphorylation at various amino acid residues, including serine1/2/19 and threonine 9/18, to regulate myosin II activity. RLC phosphorylation controls the activation and inactivation of myosin II, which was discovered in rabbit skeletal muscle [58].

The phosphorylation of RLC (serine 19 and/or threonine 18), which activates NM IIA, can be achieved by several kinds of kinases. Among the various phosphorylation sites on RLC, Ser19 controls the activation of NM IIA and the subsequent formation of bipolar filaments [59]. Phosphorylation introduces a negative charge that breaks the head–tail interaction of the inactive form [56,60,61]. Ser19 phosphorylation increases both the ATPase activity and actin-binding capacity of NM IIA [62,63]. Ser19 phosphorylation is mediated by several kinases, including MLCK, Rho-associated coiled coil-containing kinase (ROCK), and myotonic dystrophy kinase–related Cdc42-binding kinase (MRCK). ROCK enzymes target myosin phosphatase target subunit 1 (MYPT1) to increase the phosphorylation of Ser19 [64]. A conformational change from 10S to 6S via Ser19 phosphorylation is observed in all of the paralogs of NMII; however, the reverse change in conformation from 6S to 10S is paralog-dependent. Therefore, paralog dependency during the reverse conformation step is essential for controlling isoform-specific functions [65].

Apart from Ser19 phosphorylation, additional phosphorylation on Thr18 can modulate NM IIA activity. Thr18 phosphorylation stabilizes NM IIA in the 6S conformation, making the filaments more organized [66]. Therefore, upon Ser19 and Thr18 phosphorylation, the cells can exhibit three different RLC statuses: non-phosphorylated RLC, pSer19, and pSer19 + pThr18 [67]. Biphosphorylation can influence the strength of filaments; for example, filaments with pThr18/pSer19-RLC residues are present at the trailing edge [65], or in the mitotic spindle midzone, which is subjected to greater stress [68,69]. Rho kinase and protein phosphatase 1 can regulate muscle contraction and cell motility by controlling the phosphorylation state of myosin light chains (MLCs). Rho kinase phosphorylates and inhibits myosin light chain phosphatase (MLCP), a complex that includes protein phosphatase 1 as its catalytic subunit. This inhibition reduces protein phosphatase 1 activity, leading to the decreased dephosphorylation of MLC (MLC remains phosphorylated) and the sustained activation of NM IIA [28].

In addition to the phosphorylation of Ser19 and Thr18, other phosphorylation events may reduce NM IIA activity. For example, protein kinase C (PKC)-induced phosphorylation at the Ser1/2 and Thr9 positions renders NMII a poor substrate for MLCK to decrease its activity [70,71]. However, these phosphorylation events, unlike those at Ser19/Thr18, appear to have a modulatory effect on in vivo NMII activity [72]. PKC-induced phosphorylation regulates actin filament reorganization induced by platelet-derived growth factor (PDGF) in platelets [73]. Asokan et al. reported that in mesenchymal cells, local myosin IIA inactivation via the Ser1/2 phosphorylation of RLC impaired mini-filament contraction downstream of conventional PKC [74]. Kinases involved in RLC phosphorylation events are present in specific locations inside the cell. For example, MLCK is localized next to the cell membrane and activates myosin II in response to stimulation via the Ca^2+^-calmodulin complex [75]. Abl tyrosine kinase and p21-activated kinase 1 (PAK1) are involved in regulating the activity and localization of MLCK inside the cell [76,77,78]. Similarly, RhoA, which activates both ROCK and citron kinase, is localized in the middle of the cytoplasm. During the developmental stage, neuroepithelial cells, which are a type of stem cell, display the RLC phosphorylation of NMII mediated by the intracellular localization of ROCK, which is directed by Shroom 3 [79,80]. DAPK3 is a member of the death-associated protein kinase (DAPK) family. In cells undergoing apoptosis, DAPK3 localizes to the nucleus, where it phosphorylates RLC in a Ca^2+^- calmodulin-independent manner [81].

### 3.2. Role of NMHC Phosphorylation in Regulating NM IIA Activity

The first evidence of NMHC phosphorylation was observed in phagocytes, where it is required for filament assembly and disassembly [82]. Some of the common phosphorylation sites include Thr1800, Ser1803, and Ser1919 in the coiled coil domain and Ser1943 in the non-helical tail piece (Figure 4). Several kinases are involved in heavy-chain phosphorylation, including casein kinase II (CK II), PKC, the alpha kinase family, and transient receptor potential melastatin 7 (TRPM7) [83,84,85,86,87].

PKC family enzymes phosphorylate the Ser1916 and Ser1937 residues in the coiled coil domain to inhibit NMII assembly [88,89]. Phorbol esters are organic compounds that are derived from the resin of the tropical plant family Euphorbiaceae. These compounds are tumor inducers and activate PKC family members. Phorbol esters promote the phosphorylation of NM IIA at the Ser1916 residue in thrombocytes, T-lymphocytes, and mast cells [90,91]. In mast cells, the PKC-dependent phosphorylation of the NMHC is correlated with mast cell secretion via exocytosis [92]. Apart from NM IIA, PKC also phosphorylates serine in NM IIB and threonine in NM IIC [83,87].

Casein kinase II (CK II) is one of the earliest kinases discovered and has contributed immensely to the development of the human phosphoproteome [93]. It is a serine/threonine kinase that is involved in tumorigenesis, where it regulates the activity of malignant hallmarks [94]. CK II phosphorylates the Ser1943 residue of NM IIA to inhibit assembly [95]. In addition, the phosphorylation of the Ser1943 residue by CK II also controls the organization of the NM IIA assembly [84]. Beach et al. reported that in murine mammary gland epithelial cells, the induction of EMT via TGF-β signaling increases the phosphorylation of NMHC IIA at the Ser1916 and Ser1943 residues. These molecular changes are associated with alterations in filament dynamics, indicating a potential role in the cellular transitions associated with EMT [96].

Using mesenchymal stem cells, Rabb et al. reported the significance of Ser1943 as a mechanosensor. When NM IIA is phosphorylated at the Ser1943 position, stem cells exhibit a random migratory pattern on a soft matrix. However, the dephosphorylation of Ser1943 was observed when the cells were plated on a stiffer matrix. Stem cells switched to NM IIB-dependent migration when plated on a stiffer matrix [97]. Dulyaninova et al. showed that EGF stimulation results in transient Ser1943 phosphorylation of NM IIA to inhibit filament assembly [95]. Beach et al. replaced the wild-type NM IIA with a mutant NM IIA in which the Ser1943 residue cannot be phosphorylated. They observed that cells with mutant NM IIA showed an increase in assembly at lamellar regions and a decrease in cell migration [72]. Similarly, Rai et al. mutated either Ser1916 or Ser1943 to alanine (serine to alanine) and observed a decrease in NM IIA filament formation at the leading edge. When studying 3D migration, they observed that knocking down NM IIA resulted in a defect in cell invasion [98]. Transient receptor potential melastatin-subfamily member 7 (TRPM7) is a membrane protein important for the regulation of ions, such as Zn^2+^, Mg^2+^, and Ca^2+^ [99]. TRPM7 kinase can phosphorylate the heavy chain of all three paralogs of NMII. For example, TRPM7 phosphorylates Ser1803/1808 and Thr1800 in NM IIA to decrease filament assembly in vitro and reduce actomyosin formation in vivo [85,86].

### 3.3. Role of Protein Interactions in Regulating NM IIA Activity

In this section, we will focus on various protein interactions that can regulate NM IIA activity. Several proteins, including Lethal (2) giant larvae (Lgl), S100A4, and myosin binding protein H-like (MYBPHL), can interact with NM IIA to modulate its localization and filament assembly. S100A4/MTS1 is a member of the S100 family that can promote tumor development and metastasis. The S100 family of proteins regulates cell cycle events and are localized in the cytoplasm and nucleus [100]. In response to a conformational change induced by Ca^2+^, S100A4 binds to the C terminus using a hydrophobic cleft to disassemble the NM IIA filaments. Kiss et al. and Ramagopal et al. elucidated the crystal structure of the S100A4-NM IIA complex [101,102]. Although Ser1943 is not directly involved, its phosphorylation alone can disrupt the S100A4-NM IIA complex formation [84]. Li et al. showed that phagocytes lacking S400A4 display an over-assembly of NM IIA filaments, which results in reduced chemotaxis [103]. S100A4 knockdown phagocytes also exhibit a decrease in podosome formation [104]. In contrast, the overexpression of S100A4 induces metastasis by increasing cytoskeletal plasticity and migration [105]. S100-P, another member of the S100 family, can regulate NM IIA filament assembly and migration [106].

Lgl proteins are present mainly in the apical and basolateral domains of the cell. Lgl proteins are important for regulating the apico-basal polarity of cells [107]. Lgl1, a member of the Lgl family, is a tumor suppressor that interacts with NM IIA in vitro to inhibit filament formation [108,109]. Atypical protein kinase C (aPKCζ) phosphorylates Lgl1 to disrupt its interaction with NM IIA. NMIIA and aPKCζ compete to interact with Lgl1 directly [108,109]. Therefore, it is possible that Lgl1 and NM IIA might be involved in determining front–rear polarity. LIM and calponin-homology domains 1 (LIMCH1) is an actin stress fiber-associated protein that regulates NMII activity. Lin et al. reported that the N terminal region of LIMCH1 specifically binds to the head motor domain of NM IIA [110]. The knockdown of LIMCH1 enhanced migration via a decrease in actin fibers and focal adhesions in HeLa cells [110]. In lung adenocarcinoma cells, MYBPHL interacts with Rho kinase 1 to inhibit RLC phosphorylation on NM IIA [111]. MYBPHL interacts with the rod domain of the NM IIA heavy chain to inhibit the assembly and reduce the motility of lung adenocarcinoma cells. In a rat model, Zhu et al. reported that the interaction of MYBPHL with Rho kinase 1 reduces hyperplasia after injury in the carotid artery [112]. Tropomyosin 4.2 (Tpm4.2) is a tropomyosin isoform of actin-binding protein. Tpm4.2 recruits NM IIA to actin filaments during the formation of stress fibers [113].

## 4. Physiological Functions of NM IIA

The expression of NMII isoforms significantly differs in cells. Thrombocytes and the spleen express only the NM IIA isoform, whereas phagocytes express both the NM IIA isoform and the NM IIB isoform. Similarly, mature cardiomyocytes express only the NM IIB isoform, whereas non-myocytes express both the NM IIA and NM IIB isoforms. Lung tissue expresses the same amounts of NM IIA and NM IIC (40% each) and approximately 20% of NM IIB [114]. Apart from the presence of multiple isoforms, all three paralogs of NMII can co-assemble to form heterotypic filaments inside the cell. This finding implies that NMII isoforms can perform specific functions alone and co-assemble with other paralogs to carry out redundant functions [115,116,117]. The co-polymerization of NM IIA and NM IIB paralogs, coupled with their distinct turnover rates, leads to the formation of a polarized actin–NMII cytoskeleton [118]. Myosin 18A is a non-muscle II myosin that lacks ATP hydrolysis activity and can co-assemble with different NMII filaments to widen the functional range of each subtype [119].

The presence of NM IIA is important for mouse embryonic development. For example, NM IIA knock mice show poor development of the visceral endoderm [120,121]. The absence of NM IIA disrupts cell junctions, resulting in the loss of cell polarization with a disorganized endoderm [121]. NM IIA knockout embryos exhibited multinucleated cells, indicating abnormalities in cell division. In *C. elegans*, Guo et al. deleted NMY-2, which is one of the two NMHCs, and observed impaired cytokinesis in the embryo [122]. Similar results were observed by Osorio et al. when NMY-2 ATPase activity was reduced, indicating the essential role of NM IIA motor activity in cell division [123]. Mouse studies revealed that the presence of low ATPase activity in mutant NM IIA-R702 leads to abnormalities in placental development, indicating the significance of fully active NM IIA in the development process [124]. NM IIA is crucial for tissue-level developmental processes. Lienkamp et al. showed that the loss or mutation of the *MYH9* gene results in nephropathies [125]. Dysregulated NM IIA activity is linked to the pathology of glomerular diseases [126]. The regulation of NM IIA activity by Wnt signaling pathways is important for the polarization of the auditory epithelium [127]. Reduced NM IIA function may disrupt the localization of junction proteins such as vinculin [127,128]. Such disturbances may interfere with sound perception, potentially contributing to deafness, a condition linked with *MYH9*-related disorders.

Studies using various models, such as Dictyostelium and starfish [129,130,131], have demonstrated that NM IIA is necessary for mitosis [131,132]. Using both single-cell (amoebas) and multicellular organisms (fungi) and animals, studies have shown that effective cytokinesis relies on the formation of an actomyosin contractile ring enriched with NMII, which facilitates the physical separation of daughter cells [133]. Although NMII is crucial for cytokinesis, studies have shown that its motor function is not necessary in budding yeast. Some studies have reported that NMII motor activity is not a prerequisite for cell division [129,134,135,136]. However, any of the NM II isoforms can promote cell division [137]. NM IIA activity is significant for polarization and asymmetrical cell division [138,139].

Cell adhesion is dependent on integrins, which act as transmembrane linkers between the ECM and actin. Integrins, in conjunction with actin filaments and tension generated by NM IIA, facilitate cell migration [140]. Among the NMII isoforms, NM IIA shows a faster rate of ATP hydrolysis and actin filament sliding [141]. Therefore, NM IIA is well suited for rapid remodeling and actin filament assembly at the leading edge [118,142,143]. Many forms of cell polarity are marked by the asymmetric distribution of proteins, cell organelles, and cytoplasm. For example, the apical–basal polarity of epithelial cells (composed of apical and basolateral junctional complexes) is the most common form of cell polarity and is determined by intercellular adhesion contacts [144]. In polarized cells, the junction complex assembles and disassembles at the apical region where connection with actomyosin bundles maintains their integrity and stability [145].

Blebbistatin blocks the myosin head in a complex state, which has a low affinity for actin filaments, leading to a decrease in the size and density of adhesion complexes [146]. NM IIA contractile forces drive the contraction of adherent junctions, which are vital for establishing polarized epithelial tissues both in vitro and in vivo [147,148,149]. Contractile forces are essential for the formation and stability of adhesion complexes [150,151,152]. Ivanov et al. reported that the knockdown of NM IIA via small-interfering RNA (siRNA) in intestinal epithelial cells disrupts cell–cell adhesion and alters morphology [153]. Similarly, in kidney epithelial cells, the knockdown of NM IIA disrupts stable intercellular adhesions [152]. In 2D cell cultures, cells are grown on a flat surface, whereas in 3D cell culture, they grow in a three-dimensional environment, typically within a gelatinous matrix or within a solid scaffold. Cells in a 2D environment migrate by reorganizing their actomyosin cytoskeleton to form a distinct trialing and leading edge [154,155]. NMII isoforms promote 2D cell migration, and their inhibition impairs cell motility [156,157]. During migration, the leading or the front edge extends via actin polymerization, forming lamellipodia, while retrograde actin flow, driven by NM IIA, aids in cytoskeletal remodeling. The formation of focal adhesion at the leading edge is important for efficient migration [158,159,160,161,162] (Figure 5). NM IIA is primarily found at the leading or front edge, whereas NM IIB levels are high at the trailing or rear edge.

During retrograde actin flow, NM IIA levels are notable in newly formed filaments due to the faster disassembly rate from NM IIA/NM IIB bipolar filaments. Conversely, NM IIB levels are high at the cell’s rear end due to its association with older actin fibers [118]. Furthermore, while NM IIA contributes to cellular contraction and facilitates the directional migration of the cell, elevated levels of NM IIB hinder cell motility and could enhance cytoskeletal stability [118]. At the rear edge of the cell, NM IIA is essential for the movement and disassembly of focal adhesions. NM IIA promotes forward movement via the retraction of the trailing or rear edge [163,164,165,166]. The different expression levels of NMII isoforms influence the cell migration pattern.

Cells undergoing 3D migration predominantly exhibit mesenchymal and ameboid behaviors [167]. In 3D migration, cells exhibit elongated shapes with thinner protrusions. Mesenchymal migration typically involves an elongated cell morphology and relies on protease-mediated matrix remodeling for movement through narrow spaces. In contrast, ameboid migration is characterized by rounded cell shapes and a more fluid-like movement, often independent of proteolysis, and it is capable of squeezing through matrix gaps. Mesenchymal and ameboid cells exhibit variable adherence to the ECM and differ in contractile forces and the cellular localization of NM IIA [167]. Mesenchymal migration is marked by the presence of NM IIA and actin-rich protrusions at the leading edge [168]. In contrast, ameboid migration is characterized by the presence of NM IIA at the front or rear end, and ECM adhesion is not essential [169,170]. At the leading edge, there is an increase in actomyosin contractility which drives the cell forward, leading to the formation of blebs [171], whereas at the rear end, there are uropods (highly contractile structures) associated with leukocyte movement [169,172].

Plasma membrane blebbing, which is observed in both physiological and pathological states, is well documented during cell migration and apoptosis [173,174,175]. Blebs are dynamic protrusions of the cell membrane controlled by different cytoskeletal proteins [176,177]. Blebs are formed when the actomyosin cytoskeleton at the cell cortex contracts, generating high intracellular hydrostatic pressure [176,177,178]. High intracellular pressure disrupts the plasma membrane and the underlying cytoskeleton to form dynamic protrusions [178,179,180,181,182]. Bleb expansion, which lacks polymerized actin, occurs because of persistent hydrostatic pressure inside the cells [183,184]. The expanded blebs will retract gradually (Figure 6), and prior to retraction, various linker proteins, such as ezrin (which links the plasma membrane to actin), initiate actin polymerization [179]. Furthermore, bleb expansion and retraction are regulated via actomyosin cytoskeletal molecules [179]. Studies have shown that the NM IIA powers the bleb retraction process by further recruiting and accumulating on actin filaments [171,175,185,186]. 

Rho GTPases are molecular switches that regulate cytoskeleton dynamics. ROCK (a specific serine–threonine kinase) is a downstream molecule of Rho GTPase that can activate NM IIA to induce actomyosin contraction [175,181,184,185]. The ezrin is a cross-linker protein, and its recruitment to the bleb facilitates the activation of RhoA by MyoGEF (myosin-interacting guanine nucleotide exchange factor) [185]. Agents that inhibit the activities of ROCK, MLCK, and NMII (blebbistatin), along with actin-depolymerizing agents, suppress membrane blebbing [178]. Among the three NM II paralogs, NM IIA motor activity is required for bleb retraction [187]. The bleb retraction rate is correlated with NM IIA turnover on bipolar filaments [174,187]. Calcium is essential not only for NMII-mediated contraction but also for subsequent bleb formation and retraction, as shown by studies in which calcium chelators disrupted blebbing [175]. Furthermore, various calcium-dependent mechanisms, including those that activate NMII and facilitate cytoskeletal rearrangements, are known to trigger blebbing [188].

NM IIA is important for glucose-stimulated insulin secretion (GSIS) because it regulates actin and focal adhesion (FA) remodeling [189]. Arous et al. showed that the knockdown or inhibition of NM IIA decreases GSIS and alters actin stress fibers and FAs, impairing insulin secretion [189]. These findings highlight NM IIA’s central regulatory role in orchestrating the cellular processes underlying GSIS. The APPL1 adaptor protein regulates cell proliferation and mediates the adiponectin and insulin signaling pathways. Saito et al. reported that in response to mechanical stretching, APPL1 interacts with NM IIA to enhance glucose uptake [190]. NM IIA is implicated in regulating the biomechanical properties of the mouse aorta [191]. Blebbistatin or *MYH9* knockdown suppresses the activity of focal adhesion proteins such as FAK and paxillin [191].

Using motile U2-OS (human osteosarcoma) cells, Fenix et al. showed that NM IIA drives cellular movement and cytokinesis by organizing non-muscle myosin II filaments (NMII-Fs) [192]. NM IIA promotes the formation of NM IIA-F stacks through expansion and concatenation mechanisms, which are regulated by motor activity and actin filament availability. These mechanisms are crucial for cellular dynamics in both the interphase and mitotic phase [192]. *MYH9* is important in regulating DNA synthesis [193]. *MYH9* interacts directly with dNTPs with varying efficiencies [193]. The stability of *MYH9* stability is enhanced by dNTPs, and inhibition with blebbistatin or siRNA leads to decreased cell proliferation, indicating its involvement in DNA synthesis [193]. Podocytes are highly specialized cells of the glomerulus that act as barriers to prevent the filtration of plasma proteins in the urine. NM IIA activity is important for the structure and function of podocytes. The inhibition of NM IIA activity results in alterations in the podocyte structure and increased cell motility [194]. These findings suggest an association between *MYH9* mutations and glomerular pathologies [194].

NM IIA is crucial for the maturation of reticulocytes into erythrocytes [195]. NM IIA mediates axon growth and guidance in developing neurons. It is phosphorylated by the intracellular domain of EphA3, which is generated by presenilin-1/γ-secretase cleavage, leading to cytoskeleton rearrangement and facilitating axon elongation. NM IIA activity is crucial for presenilin/EphA3-dependent axon elongation, as evidenced by its ability to reverse axon retraction in presenilin-deficient neurons. Overall, NM IIA acts downstream of EphA3 cleavage to regulate axon growth, highlighting its importance in neuronal development [196].

Studies in zebrafish have shown that NMHC IIA is crucial for the formation and function of the glomerulus. The knockdown of zNMHC IIA (zebrafish NMHC IIA) induces a decrease in mesangial cells and impairs glomerular filtration [197]. NM IIA is also important for regulating cell shape changes during midbrain–hindbrain boundary (MHB) morphogenesis in zebrafish. NM IIA is specifically crucial for shortening cells at the MHB constriction (MHBC). Its activity ensures the proper angle of the MHB tissue. Furthermore, NM IIA knockdown results in an abnormal distribution of actin within MHBC cells. These findings signify the function of NM IIA in coordinating morphological changes that are essential for MHB morphogenesis [198]. NM IIA, along with gelsolin (an actin-binding protein), is integral to collagen phagocytosis by fibroblasts. It is essential for collagen binding and internalization, and its filament assembly is crucial for actin remodeling in nascent phagosomes. This assembly depends on both MLC phosphorylation and elevated intracellular calcium levels, indicating its regulation by calcium signaling. The interaction between NM IIA and gelsolin suggests a coordinated mechanism at collagen adhesion sites, where these proteins enable NM IIA filament assembly and actin remodeling in response to collagen binding, ultimately facilitating the phagocytic process [199].

## 5. *MYH9* as an Oncogene

The *MYH9* gene was initially identified as a tumor suppressor. However, subsequent studies have shown that *MYH9* is also involved in tumorigenesis and treatment resistance. In addition, it can also play a dual role as a promoter and suppressor based on cancer type (Table 2).

### 5.1. Head and Neck Squamous Cell Carcinoma

Head and neck squamous cell carcinomas (HNSCCs) originate from the squamous cells of the oropharynx and larynx. *MYH9* has been implicated in the development of radiotherapy resistance in HNSCCs. You et al. showed that the overexpression of *MYH9* enhanced cell motility via the activation of the MAPK signaling pathway, including the upregulation of ERK, P38, and JNK. MAPK pathway upregulation further activates the Nrf2 transcription factor that decreases the levels of reactive oxygen species (ROS) to promote invasion and radioresistance [200]. NEDD9 (neural precursor cell expressed, developmentally downregulated 9) belongs to the CAS (Crk-associated substrate) family of non-catalytic scaffolding proteins [237]. The activation of integrins and B cell antigen receptor (BCR) induces the hyperphosphorylation of the NEDD9 protein, which regulates tumor invasion and the chemokine response [237,238,239,240]. Although mutations in NEDD9 are rare in tumors, NEDD9 is frequently overexpressed and constitutively hyperphosphorylated, which is correlated with unfavorable clinical outcomes. NEDD9 interacts with NM IIA to increase the phosphorylation of RLC at Ser1943, contributing to matrix metalloproteinase (MMP) secretion and metastasis. The treatment of HNSCC cells with blebbistatin altered cell morphology and decreased the secretion of MMPs 2 and 9 [241].

### 5.2. Gliomas

Glioma is the most common central nervous system (CNS) tumor originating from glial cells. These tumors are highly invasive and infiltrate the surrounding brain tissue extensively. Among gliomas, glioblastoma represents the most aggressive form, while pilocytic astrocytomas are considered the least malignant type of brain tumor. Glioma cells overexpress *MYH9* and high mobility group A1 (HMGA1, a chromatin structural protein). HMGA1 activates the PI3K/Akt/c-Jun signaling pathway to upregulate the transcription of *MYH9*. High levels of *MYH9* ubiquitinate glycogen synthase kinase-3β (GSK-3β) to enhance the nuclear translocation of β-catenin that contributes to invasion and resistance to temozolomide treatment [242].

Thrombospondin 1 (THBS1) is a key player in mechanotransduction and a potential target of apatinib [205]. THBS1 interacts with *MYH9* to increase the malignancy of glioma cells. Treatment with apatinib indirectly decreased the levels of *MYH9* to inhibit the invasion and migration of glioma cells [205]. Members of the S100 family are calcium-binding proteins that are commonly upregulated in various cancers. Using 94 patients with GBM, Inukai et al. reported the significance of NM IIA in GBM tumorigenesis [206]. Immunohistochemistry data revealed an interaction between S100A4 (a member of the S100 family) and NM IIA that contributes to aggressive phenotypes in GBM. NM IIA inhibition affects the recruitment and migration of S100A4-positive GBM cells along pre-existing blood vessels. This process is mediated by the release of vascular endothelial growth factor A (VEGFA) from S100A4-positive/hypoxia-inducible factor-1α (HIF-1α)-positive tumor cells. NM IIA inhibition leads to tumor progression by enhancing pro-tumorigenic vascular functions [206].

Picariello et al. showed that NM IIA plays a critical role in inhibiting tumor invasion, and its deletion paradoxically increases tumor proliferation, a response that is influenced by the mechanical properties of the tumor environment. On softer surfaces, NM IIA deletion enhances ERK1/2 signaling, promoting proliferation, while on stiffer surfaces, it increases NF-κB activity, which also contributes to tumor progression. The findings suggest that although targeting NMIIA might be effective in preventing tumor invasion, it could inadvertently accelerate tumor growth by activating compensatory signaling pathways. Therefore, the study underscores the need for therapeutic approaches that can simultaneously inhibit both tumor invasion and proliferation, considering the mechanical context of the tumor environment to avoid unintended consequences [243].

### 5.3. Nasopharyngeal Carcinoma (NPC)

Nasopharyngeal carcinoma (NPC) is a malignant tumor linked with Epstein–Barr virus (EBV) infection. EBV-miR-BART22 is a microRNA that belongs to the EBV-encoded microRNA BART (BamHI A rightward transcript) family [207]. EBV-miR-BART22 increases *MYH9* levels via the activation of the PI3K/AKT/c-Jun pathway to induce tumor stemness, metastasis, and resistance to cisplatin. Like in glioma cells, *MYH9* promotes GSK3β protein degradation via ubiquitination to promote the nuclear translocation of β-catenin [207].

FOXO1 (a tumor suppressor), also referred to as FKHR (forkhead in rhabdomyosarcoma), belongs to the forkhead box O-class (FoxO) subfamily of transcription factors. FOXO1 regulates cell proliferation, survival, metabolism, and the oxidative stress response [208]. *MYH9* is a key player in FOXO1-mediated NPC pathogenesis and chemosensitivity to cisplatin [209]. *MYH9* interacts with FOXO1, resulting in decreased levels of *MYH9* via the activation of the PI3K/AKT/miR-133a-3p pathway. Low levels of *MYH9* lead to a reduced interaction with TNF receptor-associated factor 6 (TRAF6) and GSK3β to inhibit EMT and stemness [209,210,211]. *FNDC3B*, also referred to as *FAD104*, is a novel gene that regulates the differentiation of adipocytes and osteoblasts. The abnormal expression of *FNDC3B* is observed in hepatocellular carcinoma (HCC), acute myeloid leukemia (AML), and cervical cancer [244]. Studies have shown that FNDC3B binds and stabilizes *MYH9* to activate the Wnt/β-catenin pathway. *MYH9* is implicated in counteracting the suppressive effects of *FNDC3B* knockdown, suggesting its crucial role in mediating the oncogenic functions of *FNDC3B* [245].

### 5.4. Non-Small Cell Lung Cancer (NSCLC)

Non-small cell lung cancers encompass approximately 85–90% of all lung cancers. Katono et al. reported that high *MYH9* levels in patients with NSCLC is associated with several adverse clinicopathological features and poor prognosis [212]. Chen et al. reported that *MYH9* significantly enhances the stemness of lung cancer cells (LCCs) by regulating the expression of cancer stem cell (CSC) markers, including CD44, SOX2, Nanog, CD133, and OCT4. Furthermore, *MYH9* also contributes to NSCLC stemness via the mTOR pathway [213].

The epidermal growth factor receptor (*EGFR*) gene is overexpressed in most NSCLC cases. Patients with *EGFR* point mutations respond well to *EGFR* tyrosine kinase inhibitors (*EGFR*-TKIs), such as Gefitinib and Erlotinib. Resistance to TKIs develops due to a secondary mutation (T790M) in the intracellular tyrosine kinase domain. T790M is a missense point mutation that substitutes threonine (T) with methionine (M) at position 790 [246]. Chiu et al. reported the crucial role of *MYH9* in mediating the interaction between the drug-resistant *EGFR*-T790M mutation and β-actin. Disrupting the *EGFR*-T790M/*MYH9*/β-actin signaling axis enhances the therapeutic efficacy of CL-387,785 (a kinase inhibitor for *EGFR*-T790M) against drug-resistant NSCLC cells [214].

DT-13 is a bioactive saponin that regulates VEGF and PI3K/Akt signaling to inhibit angiogenesis and cell proliferation. It is a potent anti-metastatic compound because it inhibits the secretion of MMP-2/9 or tissue factor [247]. Under hypoxic conditions, NM IIA expression is inhibited, and its localization within lung cancer cells is altered, potentially contributing to enhanced metastatic behavior. However, treatment with DT-13 counteracts these effects by upregulating NM IIA expression and influencing its cellular distribution, thereby inhibiting metastasis [248].

Cancer-associated fibroblasts (CAFs) are activated fibroblasts that show significant heterogeneity and plasticity within the tumor microenvironment (TME) [249]. Du et al. employed a hypoxia-induced CAF model and an adipocyte model to study the impact of the TME on tumorigenesis. They observed that the conditioned medium obtained from the tumor microenvironment increased NM IIA expression and subsequent cancer cell migration [250]. Lung adenocarcinoma falls under the umbrella of NSCLC and usually originates from the mucosal glands. Microtubule-associated monooxygenase, calponin, and LIM domain containing 2 (MICAL2) is a monooxygenase that regulates cytoskeletal dynamics. Elevated levels of MICAL2 facilitate tumor development by stimulating cell proliferation and migration [251]. *MYH9* transports molecules that interact with MICAL2 in the cytoplasm. As a transporter, *MYH9* is important for MICAL2-mediated malignancy in patients with lung adenocarcinoma [215].

### 5.5. Hepatocellular Carcinoma (HCC)

Hepatocellular carcinoma (HCC) is strongly associated with viral hepatitis B/C and alcohol use [252]. *MYH9* interacts with GSK3β, adenomatous polyposis coli (APC), and axis inhibition protein 1 (AXIN1) to induce the ubiquitination of β-catenin [216]. *MYH9* forms a feedback loop with c-Jun, which transcriptionally stimulates *MYH9* expression. Furthermore, hepatitis B virus X protein (HBX) can induce the expression of *MYH9* via the modulation of GSK3β/β-catenin/c-Jun signaling [216]. Enkurin is an adaptor protein that localizes signal transduction machinery to calcium channels. In lung adenocarcinoma and colorectal cancer, Enkurin functions as a tumor suppressor [253,254]. In HCC, Enkurin interacts with *MYH9* to inhibit the nuclear translocation of β-catenin. This interaction decreases the levels of both c-Jun and *MYH9*. Reduced levels of *MYH9* inhibit cell proliferation and resistance to sorafenib [217].

In lung adenocarcinoma, *MYH9* stabilizes c-Myc via deubiquitination with the help of the deubiquitinating enzyme ubiquitin-specific peptidase 7 (USP7) to promote cell cycle progression and EMT signaling [255]. NMHC IIA plays a critical role in HCC metastasis, where its phosphorylation at Ser1943 is dynamically regulated by bradykinin, leading to increased cell migration and invasion. NMHC IIA is integral to cytoskeletal dynamics and cellular contractility. The activation of TRPM7 by bradykinin results in the phosphorylation of the myosin IIA heavy chain and subsequent modulation of cell and focal adhesion dynamics [256,257]. Nucleosome assembly protein 1-like 5 (NAP1L5) is a histone chaperone that acts as a tumor suppressor. In HCC, NAP1L5 downregulates *MYH9* which, in turn, inhibits PI3K/AKT/mTOR signaling, leading to its tumor-suppressive effects [258].

### 5.6. Pancreatic Cancer (PC)

Pancreatic cancer (PC) originates from pancreatic duct cells. Zhou et al. found that NM IIA expression is significantly high in PC tissue compared with normal tissue. NM II regulates β-catenin transcriptional activity to promote tumorigenesis in PC cells both in vitro and in vivo [221]. Pancreatic acinar cell carcinoma (PACC) is an uncommon malignant tumor characterized by cells resembling acinar cells and exhibiting exocrine enzyme production. It constitutes approximately 1–2% of all pancreatic neoplasms and is considered a high-grade malignancy [259]. 4H12 is a monoclonal antibody generated against Faraz-ICR, a PACC cell line, that specifically targets *MYH9*. Using immunohistochemical staining, the authors observed high *MYH9* expression in acinar cell tumors, the source of Faraz-ICR cells. The 4H12 mAb suppressed Faraz-ICR cell proliferation in a dose-dependent manner [260].

### 5.7. Esophageal Cancer

Esophageal cancer is classified into squamous cell carcinoma (SCC) and adenocarcinoma (ADCA). NM IIA expression is significantly high in patients with esophageal squamous cell carcinoma (ESCC) and is associated with lymph node metastasis, an advanced tumor stage, and shorter overall survival. Low NM IIA expression in ESCC cells leads to increased cell–matrix adhesion and decreased cell migration [218]. Yang et al. used genomic sequencing data from 104 patients with ESCC and found that *MYH9* levels were elevated in patients with ESCC with lymph node metastasis. A PCR array analysis revealed that *MYH9* knockdown altered angiogenesis and EMT gene expression [219]. Protein tyrosine phosphatase 1B (PTP1B) dephosphorylates *MYH9* at tyrosine 1408 (Y1408), leading to the upregulation of *EGFR* expression and subsequent proliferation [220].

### 5.8. Gastric Cancer (GC)

Gastric or stomach cancer (GC) originates in the stomach and is one of the most prevalent cancers worldwide. In GC, *MYH9* plays a critical role in the context of peritoneal metastasis. Nuclear *MYH9* promotes the transcription of the β-catenin gene (*CTNNB1*) to activate the Wnt/β-catenin pathway, which, in turn, confers resistance to the anoikis form of cell death. The inhibition of *MYH9* phosphorylation, particularly at Ser1943, by compounds such as staurosporine, inhibits *CTNNB1* gene transcription [222]. The downregulation of NM IIA using siRNA leads to the downregulation of the JNK signaling pathway, which promotes GC cell migration and invasion [223]. *MYH9* plays an essential role as a downstream effector of the Enkurin pathway. Enkurin recruits the E3 ubiquitin ligase F-Box and WD Repeat Domain Containing 7 (FBXW7) enzyme to downregulate *MYH9* levels [261]. Liang et al. reported that the miRNA let-7f (a member of the let-7 family of microRNAs) functions as a tumor suppressor by targeting *MYH9*. The overexpression of the miRNA let-7f downregulates *MYH9* levels to decrease migration and invasion [262].

Hepatocellular carcinoma upregulated long non-coding RNA (HULC) is an oncogenic long non-coding RNA (lncRNA) that acts like a molecular sponge for miR-9-5p mRNA to regulate *MYH9* expression. The dysregulation of the HULC/miR-9-5p/*MYH9* axis contributes to GC progression. The knockdown of HULC inhibits tumor growth, which highlights the potential significance of targeting the HULC/miR-9-5p/*MYH9* axis for GC treatment [263]. Similarly, circSLAMF6, a circular RNA, upregulates *MYH9* expression by sponging miR-204-5p. The knockdown of *MYH9* attenuates glycolysis and invasion, indicating its role in promoting cancer hallmarks under hypoxia. Moreover, in vivo experiments have shown that circSLAMF6 deficiency inhibits tumor growth [264].

miR-647, which is downregulated in GC, suppresses in vitro migration and invasion. miR-647 targets the serum response factor (SRF) mRNA, which then upregulates *MYH9* transcription. Studies in orthotropic GC models have demonstrated that the overexpression of miR-647 inhibits metastasis, whereas increased SRF expression reverses this effect [265]. *MYH9* is consistently overexpressed in the peritoneal metastasis of advanced GC. The siRNA-mediated knockdown of *MYH9* alters the levels of EMT marker genes. Interference with S100A4, an oncogene implicated in metastasis, leads to *MYH9* downregulation and inactivation of the Smad pathway, which participates in the EMT process. *MYH9* is downstream of the S100A4 pathway and promotes EMT and metastasis [266].

### 5.9. Colorectal Cancer (CRC)

Colorectal cancer (CRC) ranks as the second most common cause of cancer-related deaths globally. Most CRC cases are adenocarcinomas, comprising more than 90% of cases, with less common subtypes including adenosquamous, spindle, squamous, and undifferentiated carcinomas. NM IIA activates the AMPK/mTOR pathway to promote tumorigenesis. NM IIA knockdown downregulates the expression of CD44 and CD133, indicating a role in regulating stemness. NM IIA protects CRC cells from 5-fluorouracil (5-FU)-induced apoptosis and prevents the inhibition of proliferation through the activation of the AMPK/mTOR pathway [267]. *MYH9* promotes tumorigenesis in CRC by modulating MAPK/AKT signaling and influencing EMT, leading to aggressive cancer behavior [224].

Zhong et al. reported that *MYH9* promotes CRC metastasis by interacting with autophagy-related protein 9B (ATG9B). *MYH9* binds to ATG9B, enhancing its stability by preventing its ubiquitin-mediated degradation. The interaction between ATG9B and *MYH9* stabilized both proteins by reducing their binding to the E3 ubiquitin ligase STIP1 Homology and U-Box Containing Protein 1 (STUB1), which would otherwise target both proteins for degradation. During cell invasion, *MYH9* promotes the transport of ATG9B to the cell edge where it enhances focal adhesion and metastasis by activating integrin β1 [225]. LIM kinase 1 (LIMK1) upregulates both *MYH9* and actinin-4 (ACTN4) to increase the aggressive phenotype associated with CRC [226]. *TIMELESS (TIM)*, a circadian gene with evolutionary conservation, encodes a Timeless protein. It is important for facilitating effective DNA replication and maintaining genome integrity. *TIMELESS* interacts with *MYH9* to promote its nuclear translocation and the subsequent activation of the β-catenin pathway, which promotes CRC tumorigenesis [268].

*MYH9* binds to polymorphic adenoma-like protein 2 (PLAGL2), a downstream molecule regulated by miR-214-3p, to promote tumor development [269]. Circular *MYH9* (circ*MYH9*), an intron-derived circular RNA, plays a crucial role in CRC by promoting serine/glycine metabolism and reducing ROS generation to facilitate tumor growth [270]. In response to amino acid deprivation, circ*MYH9* expression is induced by hypoxia-inducible factor 1 subunit-α (HIF1α), which exacerbates CRC progression. Mouse studies have shown that circ*MYH9* overexpression promotes chemically induced carcinogenesis by suppressing p53 [270].

### 5.10. Breast Cancer

Breast cancer is among the most prevalent cancers in women, both in terms of newly diagnosed cases and mortality rates. S100A4 belongs to the S100 family and has been implicated in promoting metastasis in various tumors. High levels of S100A4 are correlated with a more aggressive cancer phenotype and poorer patient outcomes. S100A4 interacts with *MYH9* to inhibit NM IIA filament formation and promote metastasis [271,272]. However, Wang et al. found that S100A1, another member of the S100 family, can inhibit the metastatic function of S100A4 [271]. S100P (member of S100 family) is upregulated in various cancer types [273]. S100P can form a heterodimer with S100A1 that weakens the binding affinity of S100P for *MYH9* [274]. S100P interacts with NM IIA to promote cell migration. However, in cells that do not naturally express NM IIA or have low basal levels of NM IIA, S100P interacts with α, β-tubulin to promote cell migration [275].

Circular EIF6 (circ-EIF6) is an oncogenic circular RNA that encodes the novel peptide EIF6-224 [276]. EIF6-224 increases the level of *MYH9* by inhibiting its degradation via the ubiquitin proteasome pathway. High *MYH9* activates the Wnt/β-catenin signaling pathway to promote the metastasis of TNBC cells [276]. Furthermore, in TNBC cells, myosin IIA along with integrin β1 and MLCK contribute to adaptive resistance development against MEK inhibitors by mediating crosstalk between the Ras-ERK and PI3K-AKT signaling pathways [227]. In contrast, tissue factor pathway inhibitor 2 (TFPI-2), a tumor suppressor, binds to *MYH9* and actinin-4 to inhibit the proliferation of breast cancer cells [277]. NM IIA modulates the biophysical characteristics of the ECM during cancer progression. In TNBC cells, the loss of NM IIA results in changes in the collagen stiffness of the constructs, which highlights its importance in efficient matrix remodeling during metastasis [228]. Derycke et al. highlighted the significance of NM IIA in breast cancer cell invasion into surrounding tissue in vitro [278].

Interferon-γ (IFN-γ) regulates the interaction between NM IIA and interferon-stimulated gene 15 (*ISG15*) in the cytoplasm of breast cancer cells [279]. Heat shock protein (HSP) 47 interacts with NM IIA via the unfolded protein response transducer IRE1α (encoded by the endoplasmic reticulum to nucleus signaling 1 (*ERN1*) gene) to enhance the metastatic potential of breast cancer cells [280]. Hepatitis B X-interacting protein (HBXIP) is an oncoprotein that interacts with the assembly-competent domain (ACD) of NMHC IIA. The phosphorylation of Ser1916 on the heavy chain of NM IIA by PKCβII further strengthens this interaction [281]. In HER2+ breast cancer cells, NM IIA levels increase upon treatment with the pan-HER inhibitor neratinib, suggesting its role in tumor growth. NM IIA modulation affects HER3 levels and downstream signaling, impacting cell growth and invasion [282]. DT-13 inhibits NM IIA and tumor microenvironment (TME)-induced breast cancer cell migration [229].

### 5.11. Renal Cell Carcinomas (RCCs)

Renal cell carcinomas (RCCs) are malignant tumors that originate from the renal cortex. Nuclear C-X-C motif chemokine receptor 4 (CXCR4) plays a key role in RCCs. NM IIA promotes RCC metastasis by facilitating the nuclear translocation of CXCR4 [230]. Clear cell renal cell carcinoma (ccRCC) is a predominant form of RCC, accounting for more than 90% of cases. *MYH9* is upregulated in ccRCC and plays an important role in its development. Single-cell RNA sequencing (scRNA-seq) data show that *MYH9* activates the AKT signaling pathway to promote ccRCC development. The *MYH9*/AKT signaling axis also confers resistance to sunitinib therapy in ccRCC [283].

### 5.12. Prostate Cancer

Prostate cancer is the second most common cause of death among men in the United States [231]. Samples from patients with prostate cancer show a high expression of human tubulin beta class IVa (TUBB4A), a member of the β-tubulin family. Cell migration, especially in narrow spaces, can induce genomic stress in prostate cancer cells. *MYH9* interacts with TUBB4A to maintain genomic stability by protecting the nucleus against severe DNA damage [232].

### 5.13. Osteosarcoma

Osteosarcoma (OS) is a malignant bone tumor that originates from mesenchymal cells. MRPL23-AS1 is a lncRNA whose expression is increased in OS cell lines and tissues. Elevated levels of the lncRNA MRPL23-AS1 result in the increased expression of *MYH9* via the competitive inhibition of miR-30b. Upregulated *MYH9* promotes metastasis via the activation of β-catenin signaling [233].

### 5.14. Papillary Thyroid Carcinoma (PTC)

Papillary thyroid carcinoma (PTC) is an endocrine cancer that accounts for approximately 90% of all thyroid cancer cases. *MYH9* enhances the stability of the cytokine receptor-like factor 1 (CRLF1) protein, which activates the ERK pathway. This activation leads to the upregulation of ETS variant transcription factor 4 (ETV4), which subsequently promotes the secretion of matrix metalloproteinase 1 (MMP1), facilitating PTC cell proliferation and metastasis [234].

In differentiated thyroid carcinoma (DTC), *MYH9* contributes to radioresistance. Circular NIMA-related kinase 6 (NEK6) (circ_NEK6) promotes resistance to iodine 131(^131^I)-based radiotherapy by regulating *MYH9* expression through the miR-370-3p/*MYH9* axis. Upregulated *MYH9* serves as a downstream effector of circ_NEK6 to promote proliferation and radioresistance [284]. *MYH9* inhibits the activity of the bidirectional promoter shared by Forkhead Box E1 (FOXE1) and the lncRNA gene papillary thyroid carcinoma susceptibility candidate 2 (*PTCSC2*) [235].

### 5.15. Acute Myeloid Leukemia (AML)

Acute myeloid leukemia (AML) is characterized by impaired erythropoiesis and bone marrow dysfunction. AML cells show enhanced actomyosin contractility due to the increased expression of NM IIA and the hyperphosphorylation of the RLC [236].

### 5.16. Diffuse Large B Cell Lymphoma (DLBCL)

Diffuse large B cell lymphoma (DLBCL) is the most common form of high-grade non-Hodgkin’s lymphoma in the United States. Glycoprotein prostaglandin D2 synthase (PTGDS), a glycoprotein belonging to the lipocalin superfamily, has dual functions in prostaglandin metabolism and lipid transportation. PTGDS participates in multiple cellular processes, including the initiation of solid tumor formation. *MYH9* interacts with PTGDS to promote the development of DLBCL via the activation of the Wnt-β-catenin-STAT3 pathway [285].

## 6. The Role of NM IIA as a Tumor Suppressor in Mice

### 6.1. Squamous Cell Carcinoma of the Skin

Squamous cell carcinoma is a common skin malignancy that has a precursor lesion and actinic keratosis. Schramek et al. were the first to report that suppressing *MYH9* expression via knockdown or RNA interference led to the emergence of an aggressive form of HNSCC and skin squamous cell carcinoma in mice predisposed to tumors. In patients with squamous cell carcinoma, Schramek et al. found that the downregulation of the *MYH9* gene is associated with poor survival. In human and mouse keratinocytes, *MYH9* promotes the nuclear accumulation of p53 to enhance its post-transcriptional stability [202]. In mouse tongue squamous cell carcinoma, *MYH9* is essential for maintaining mitotic stability during karyokinesis, and its deletion promotes carcinoma progression, indicating its role as a potential tumor suppressor. However, contrary to an earlier study by Schramek et al. who found that *MYH9* stabilizes p53, this study suggests that the tumor suppressor activity of *MYH9* is due to its role in maintaining mitotic stability during nuclear division [203].

### 6.2. Melanoma

Melanoma arises from the malignant transformation of melanocytes and accounts for approximately 4% of skin cancers. Despite its relatively low prevalence, melanoma is responsible for most skin cancer-related deaths, accounting for approximately 80% of mortality. *MYH9* knockdown or silencing in melanoma cells promoted tumor growth and metastasis. The modulation of *MYH9* altered the expression of oncogenes involved in EMT and the Erk signaling pathway. Therefore, in a melanoma mouse model, *MYH9* may act as a potential tumor suppressor [204]. The deletion of both NM IIA and NM IIB isoforms in mammary epithelium cells results in increased proliferation in 3D culture, even under conditions that do not typically support tissue growth [286].

## 7. Dual Role of NM IIA in Humans

Coaxum et al. reported the tumor suppressor activity of *MYH9* in human HNSCC. Low levels of *MYH9* are associated with decreased survival in patients with HNSCC, particularly those harboring the low-risk mutant TP53 (mutp53). The inhibition of NM IIA increases the invasive ability of cells harboring wild-type p53 (wtp53), indicating its role in suppressing invasive behavior. However, in cells carrying mutant p53, *MYH9* expression did not correlate with disease prognosis. NM II inhibition not only decreased the expression of p53 target genes but also the nuclear localization of wtp53. This may be due to the significance of *MYH9* for the nuclear retention and function of wtp53 [201]. Coaxum and their colleagues focused on the interaction between *MYH9* and p53 without diving into the functional significance of *MYH9* in HNSCC [201]. In contrast to the results of Coaxum et al., You and colleagues reported the tumor-promoting role of *MYH9* in HNSCC. *MYH9* activates pan-MAPK signaling molecules to modulate the ECM and promote radioresistance [200]. The above studies indicate that *MYH9* acts as a tumor suppressor in mouse HNSCC.

## 8. Role of NM IIA in Other Pathological Conditions

NM IIA-associated diseases can occur due to the presence of mutations, splicing errors, or the dysregulation of NM IIA function [28,287]. These mutations are commonly observed in the heavy chain of NM IIA and are referred to as *MYH9*-related diseases (*MYH9*-RD) (Figure 7). *MYH9*-RD (autosomal dominant) is characterized by the presence of congenital thrombocytopenia with giant thrombocytes and leucocyte inclusions. *MYH9*-RD results in a plethora of syndromes, such as May–Hegglin anomaly (MHA), Sebastian syndrome (SBS), Fechtner syndrome (FTNS), and Epstein syndrome (EPS) [288,289,290,291]. Additional symptoms may include hearing loss, early-onset cataracts, increased liver enzyme levels, and progressive kidney damage that may ultimately result in end-stage renal disease (ESRD) [292].

The molecular mechanism underlying *MYH9*-RD involves specific mutations in the *MYH9* gene that disrupt filament assembly, alter ATPase activity, and impair the actin-binding ability of NM IIA [293]. Structural and functional defects in NM IIA lead to the development of macrothrombocytopenia, hearing loss, renal failure, and cataracts [293]. Pecci et al. analyzed 108 patients with *MYH9*-RD from 50 unrelated families and observed that patients with mutations in the heavy chain motor domain had severe thrombocytopenia, nephritis, and deafness [294].

Various genetic alterations, including nonsense and frameshift mutations of the heavy chain, duplications, and in-frame deletions, can lead to *MYH9*-RD [295,296,297,298]. NM IIA is the only paralog expressed in platelets. It is important for the differentiation and maturation of megakaryocytes, which are precursors of mature thrombocytes. The current model of platelet formation establishes that megakaryocytes terminally differentiate and release platelets from cytoplasmic extensions called proplatelets. Mutations such as R702C, D1424N, and E1841K disrupt proplatelet formation, leading to the formation of giant proplatelets. *MYH9* mutations impair the migration of megakaryocytes within the bone marrow, preventing proper movement toward the vasculature for platelet release [299,300,301,302,303]. Mutations can also alter the cytoskeleton dynamics of platelets [304]. Platelet adhesion is impacted by a decrease in glycoprotein 1b/IX/V in large platelets in patients with *MYH9*-RD, which could contribute to their bleeding tendencies [305].

May–Hegglin anomaly (MHA), Sebastian syndrome (SBS), Fechtner syndrome (FTNS), and Epstein syndrome (EPS) are autosomal dominant diseases associated with macrothrombocytopenia, sensorineural hearing loss, cataracts, and nephritis. Heath et al. evaluated 27 patients and found *MYH9* mutations in 74%. R702C and R702H mutations are associated with FTNS, EPS, and Alport syndrome with macrothrombocytopenia (APSM) [306]. Seri et al. evaluated 19 families and reported an abnormal NMHC IIA distribution in leukocytes. A clinical evaluation revealed high-tone hearing loss in 83% of patients and cataracts in 23% of patients initially diagnosed with MHA or SBS [307]. Several studies have reported extra hemorrhagic symptoms in patients with *MYH9*-RD. Lalwani et al. reported the presence of the guanine (G) > adenine (A) mutation at nucleotide 2114, leading to an R705H missense mutation, which is associated with autosomal dominant hearing impairment [290]. *MYH9* mutations can lead to the development of nephropathy [308,309]. Elevated liver enzymes are frequently found in patients with *MYH9*-RD but do not appear to cause significant structural damage [310].

Data from 36 studies using whole-exome sequencing (WES) and whole-genome sequencing (WGS) revealed 17,104 de novo mutations across four psychiatric disorders: autism spectrum disorder, epileptic encephalopathy, intellectual disability, and schizophrenia. *MYH9* mutations were observed in patients with schizophrenia, intellectual disability, and autism [311]. Brain tissues from patients with schizophrenia show elevated levels of phosphorylated RLCs. In amyotrophic lateral sclerosis (ALS), miR-155, which is involved in the regulation of NM II, is upregulated, contributing to disease progression. Similarly, in multiple sclerosis (MS), miR-155 promotes inflammatory responses [312,313,314].

Aberrant NM IIA expression or activity is associated with microbial infections. The genetic ablation of NM II using shRNA or pharmacological inhibition impairs the dissemination of microbes [315]. NM IIA is critical for the host response to bacterial infection, particularly *Listeria monocytogenes* infection. The phosphorylation of NM IIA at the tyrosine-158 residue limits the intracellular level of *L. monocytogenes*. This phosphorylation is mediated via the tyrosine kinase Src and occurs near the ATP-binding side of the motor head domain [316]. Kaposi’s sarcoma-associated herpesvirus (KSHV), also known as human herpesvirus-8 (HHV-8), targets NM IIA to induce its phosphorylation, which results in blebbing and macropinocytosis during viral infection [317]. NM IIA is important in the defense against bacterial pore-forming toxins (PFTs), such as listeriolysin O (LLO). Following PFT-induced damage, NM IIA coordinates with the ER chaperone Gp96 to regulate cytoskeletal dynamics, membrane blebbing, and cell migration [318,319]. Similarly, perfringolysin O (PFO), which is a pore-forming toxin produced by *Clostridium perfringens*, induces a significant reorganization of the actomyosin cytoskeleton [188].

## 9. Targeting *MYH9*

The small molecule myosin inhibitor blebbistatin was identified via a screen targeting NM IIA. It acts as a non-competitive inhibitor and stabilizes the metastable state of myosin which precedes the force-generating step catalyzed by ATP hydrolysis [320]. Limouze et al. evaluated the specificity and potency of blebbistatin across various members of the myosin superfamily [321]. With an IC_50_ value between 0.5 and 5 μM, it showed a potent inhibition of various striated muscle myosin and vertebrates NM IIA and NM IIB. It showed weak inhibitory activity against smooth muscle myosin with an IC_50_ value of 80 μM [321]. Despite its broad inhibitory effect across myosin isoforms, it has several limitations, including fluorescence, poor water solubility, cytotoxicity, and sensitivity to photodegradation [322].

Several drugs that modulate *MYH9* activity are listed in Table 3. Cinobufotalin, a natural compound isolated from toad venom, has cardiotonic, diuretic, and potential cytotoxic effects. In nasopharyngeal carcinoma, cinobufotalin increases the levels of MAP2K4 to reduce cancer stemness, EMT, and resistance to cisplatin [207]. Furthermore, in HCC and lung cancer, cinobufotalin exerts an anti-tumor action by decreasing the levels of *MYH9* [217,255]. Apatinib is a small molecule TKI that inhibits angiogenesis by blocking the vascular endothelial growth factor (VEGF)-induced phosphorylation of the VEGF Receptor-2 (*VEGFR*2) pathway. In glioma cells, apatinib targets thrombospondin 1 (THBS1) to inhibit malignancy [205]. Disulfiram is an FDA-approved drug for treating alcoholism. Robinson et al. screened approximately 3185 compounds and identified disulfiram (IC_50_ value of 300 nM) as a potent inhibitor of growth in TNBC cells. Disulfiram directly targets *MYH9* and the Ras GTPase-activating-like protein IQGAP1 to inhibit growth. The combination of disulfiram and doxorubicin enhanced the killing of cancer stem cells by inducing cellular senescence [323]. C_70_-EDA is an aminated fullerene derivative that binds to the C terminal tail domain and modulates the subcellular distribution of *MYH9* to inhibit the NM IIA filament [324]. J13 is a small molecule that disrupts the interaction of *MYH9* with actin to increase mitochondrial fission. This study also revealed that heat shock protein A9 (HSPA9), a key player affected by *MYH9*-actin dysfunction, is crucial for modulating mitochondrial fission [325].

In gastric cancer, staurosporine decreases the activity of casein kinase II, leading to the reduced phosphorylation of *MYH9* at the Ser1943 residue. This results in the downregulation of *CTNNB1* gene transcription with the decreased activation of β-catenin signaling [222]. Homoharringtonine (HHT) is a natural alkaloid that upregulates *MYH9* expression in acute and chronic myeloid leukemia. High *MYH9* levels enhance the sensitivity of leukemia cell lines to HHT-induced cell death [326]. *MYH9* is also involved in cisplatin-induced kidney injury. *MYH9* interacts with upregulated apurinic/apyrimidinic endonuclease 2 (APE2) in proximal tubule cells to induce mitochondrial damage and kidney injury [341].

### Targeting the ROCK-Myosin II Signaling Pathway

Rho GTPases are molecular switches that regulate cytoskeleton dynamics [342]. Rho GTPase enzymes are normally inactive when bound to GDP; however, upon activation, they are bound to GTP to activate downstream signaling molecules such as Rho-associated protein kinases 1 and 2 (ROCK1 and ROCK2) [342,343]. ROCK enzymes are serine/threonine AGC kinases that can be activated by other mechanisms, including mechanical stress, plasma membrane binding [344], and proteolytic cleavage by caspases and granzyme B [343]. They have a molecular weight of approximately 160 kDa and differ in their expression levels across tissues [345]. These enzymes participate in multiple processes, including cell migration, differentiation, apoptosis, and immune responses [346].

Myosin light chain 2 (MLC2) is a key player in the myosin II pathway. ROCKs phosphorylate MLC2, which then binds with the heavy chain of myosin (MHC) to promote actomyosin contraction [347,348,349,350]. Myosin phosphatase (MYPT) dephosphorylates myosin II and reduces its motor activity. However, ROCK can phosphorylate MYPT to reduce its phosphatase activity, which ultimately enhances myosin II motor activity [343]. LIM kinases are actin-binding kinases that are important regulators of actin polymerization and microtubule assembly. ROCK can activate LIM kinases, which prevent the disassembly of actin filaments [351]. Other poorly characterized downstream effectors of ROCK enzymes include elongation factor-1α (EF-1α), adducin, and calponin [352].

Studies have shown that some tumors proliferate and undergo transformation via the Rho–ROCK–myosin II pathway [353,354]. However, other tumors may use the ROCK–myosin II pathway for migration and EMT [347,355]. The ROCK–myosin II pathway also contributes to treatment resistance [349], the modulation of immune responses [350], and ECM regulation [356,357]. In cancer-associated fibroblasts (CAFs), ROCK enzymes regulate actomyosin dynamics to promote cancer cell migration through the ECM [358]. Wong et al. reported that C-C motif chemokine ligand 2 (CCL2) secretion by pericytes increases tumor growth via the activation of the ROCK2–myosin II pathway [359]. The ROCK–myosin II signaling pathway enables ATPase-driven actomyosin contraction. ROCK inhibitors inhibit ROCK activity and modify the ROCK–myosin II pathway, which can be beneficial in treating diseases such as glaucoma, cancer, renal failure, and fibrosis. Since ROCK enzymes regulate axon regeneration, ROCK inhibitors may offer some advantages in treating neurological disorders [327,360].

Currently, four different ROCK inhibitors are clinically approved. Fasudil is approved to treat symptoms of cerebral ischemia [327]. The overactivation of the ROCK signaling pathway contributes to inflammation and neuronal damage in neurodegenerative diseases [361]. Fasudil was the first clinically used ROCK inhibitor for neurodegenerative diseases. Fasudil repressed neuroinflammation and decreased the secretion of inflammatory factors by inhibiting CD4+ T-cell differentiation and phagocytes, thus restoring the balance of immune cells in vitro [362,363]. Fasudil also has a neuroprotective effect by reducing neuronal apoptosis due to the overexpression of ROCK in primary mouse hippocampal neurons [364,365,366]. The combination of fasudil with gemcitabine increased the uptake and efficacy of gemcitabine in PDAC [367]. The early treatment of pancreatic cancer with fasudil remodeled the metastatic niche by increasing blood vessel formation and reducing collagen deposition [368]. Hypoxia, a hallmark of many tumors, is involved in metastatic transition. The fasudil prodrug, which remains inactive under normoxia but becomes active under hypoxic conditions, can inhibit ROCK activity without inducing hypotension [369]. However, the use of fasudil is limited due to its toxicity, non-specificity for other kinases, and poor oral bioavailability.

In Japan, ripasudil is approved for treating patients with open-angle glaucoma and ocular hypertension [328]. Ripasudil, when combined with a programmed death-1/programmed death ligand-1 (PD-1/PD-L1) checkpoint inhibitor, recruits CD8+ T cells to increase the immune response in patients with uveal melanoma [370]. Netarsudil is a pan-ROCK inhibitor approved in both the EU and the USA for patients with open-angle glaucoma with raised intraocular pressure [329]. Belumosudil is a specific ROCK2 inhibitor approved in the USA for patients with chronic graft versus host disease (GVHD) who fail to respond to first-line therapy [330,331].

Y27632 is an orally active ROCK inhibitor that targets other kinases, including PKA and PKC [332]. Y27632 increases the effectiveness of doxorubicin in vivo and reduces the tumor size by inducing an anti-tumor immune response [371]. In multiple myeloma, Federico et al. used Y27632 to improve bortezomib efficacy via the disruption of the bone marrow microenvironment [372]. In melanoma, tumors resistant to BRAF inhibitors show a strong activation of Rho/ROCK signaling, which is linked to increased dedifferentiation. A machine learning analysis using Y27632 or fasudil suggested that poorly differentiated tumors are more sensitive to ROCK inhibition. Furthermore, both Y27632 and fasudil were able to restore the efficacy of BRAF inhibitors in resistant melanoma cells [373]. ROCK stabilizes the PD-L1 protein via moesin (MSN) phosphorylation, which is linked to poor prognosis in patients with breast cancer [374].

Studies have shown that silencing ROCK1 via genetic modulation or inhibition with GSK269962A (a selective ROCK1 inhibitor) enhances the effectiveness of BRAF and MAPK inhibitors against BRAF and NRAS mutant melanoma cells [333]. Furthermore, in vivo studies using a combination of ROCK inhibitors (GSK269962A or fasudil) with MEK or ERK inhibitors suppressed NRAS mutant melanoma growth [375]. The combination of *EGFR* and ROCK inhibitors has a synergistic effect and significantly reduces the growth of TNBC cells via cell cycle arrest [376]. A mass spectrometry-based proteomic approach revealed the mechanisms involved in the co-inhibition of *EGFR* and ROCK. *EGFR* inhibition alone induces autophagy, but combining it with GSK269962A impairs autophagosome clearance [377]. AT13148 is a ROCK/AKT inhibitor that has shown positive results in preclinical cancer models but has failed in clinical trials. A poor pharmacokinetic profile combined with a narrow therapeutic index led to the termination of the trial [334].

BDP5290 is a potent inhibitor of MRCK (myotonic dystrophy kinase-related CDC42-binding kinase), which is important for actomyosin contractility and metastasis. BDP5290 binds to the kinase domain of MRCK and blocks MLC phosphorylation to inhibit the invasion of breast cancer cells [335]. Using an RNAi screen, Castoreno et al. identified Rhodblock 6 as a Rho kinase inhibitor. It disrupts the correct localization of phosphorylated MLC during cell division [336]. RKI-1447 binds to the ATP-binding site of the ROCK enzyme to inhibit the binding of MLC-2 and MYPT-1 without affecting other kinases, such as AKT, MEK, and S6 kinase. It suppresses ROCK-mediated cytoskeletal reorganization, specifically inhibiting actin stress fiber formation [337]. Similarly, RKI-18 is also a potent Rho-kinase inhibitor with significant anti-tumor properties. It inhibits the phosphorylation of ROCK and MLC2, leading to diminished lamellipodia and filopodium formation in breast cancer cells [338].

Novel thiosemicarbazone iron chelators, such as di-2-pyridylketone 4,4-dimethyl-3-thiosemicarbazone, have shown potential in cancer therapy by inhibiting metastasis. These chelators work by inducing cellular iron depletion, which upregulates the expression of metastasis suppressor N-Myc downstream regulated 1 (NDRG1). NDRG1 subsequently inhibits the ROCK1/p-MLC2 pathway to reduce migration and stress fiber formation [339]. Combretastatin (CA-4) and its analog CA-432 are potent anti-tumor agents that activate the RhoA/ROCK signaling pathway to increase the phosphorylation of MLC. Furthermore, these agents inhibit T-cell migration and chemotaxis by destabilizing microtubules, decreasing tubulin acetylation and microtubule stability [340].

## 10. Conclusion and Perspectives

NM IIA is a vital component of the actomyosin cytoskeleton that controls cell division and migration. It is important in early embryonic development for the formation of functional visceral endoderm and the maintenance of cell polarity. NM IIA is expressed not only in the cytoplasm but also in the cell membrane and nucleus. NM IIA activity is regulated at the levels of protein folding, filament assembly, and Mg^2+^-ATPase motor activity. NM IIA acts as a tumor promoter or a suppressor based on the tumor type and molecular environment. For example, NM IIA is reported to be a tumor suppressor in melanoma and squamous cell carcinoma. However, the main drawback of these studies is the use of mouse models instead of human cell lines or patient data. In contrast, numerous studies using human cancer cells and clinical data have reported the tumor-promoting role of NM IIA.

NM IIA also plays a significant role in promoting cancer cell migration, invasion, and metastasis; maintaining stemness; and regulating metabolism. Furthermore, it also inhibits apoptosis and autophagy and contributes to the development of resistance to chemoradiotherapy and targeted therapy by regulating the MAPK, PI3K/Akt, and Wnt/β-catenin signaling pathways. Therefore, because of the dual nature (as a tumor promoter and suppressor) and complexity of NM IIA’s function, it is necessary to further evaluate its molecular interactions and regulatory mechanisms using new technologies. Apart from its role in cancer, NM IIA is also involved in other pathological conditions such as renal disease, severe thrombocytopenia, and various neuropsychiatric and neuroinflammatory disorders. In this review, we discussed the significant role of NM IIA in tumor development and the pathogenesis of other diseases, underscoring its importance as a key therapeutic target. However, there are currently very few drugs or chemical agents targeting NM IIA, and the majority of them are still being studied in preclinical models. For example, blebbistatin, which inhibits NM IIA motor activity, is poorly water soluble, cytotoxic, phototoxic, and unstable with limited specificity. The mechanism of action and efficacy of some drugs, such as disulfiram and C_70_ -EDA, are not well understood.

ROCK enzymes are important for the regulation and activation of NM IIA. Fasudil, ripasudil, netarsudil, and nelumosudil are currently approved for targeting the ROCK pathway. Many preclinical cancer studies have demonstrated the significance of ROCK inhibitors, but there is one clinical trial targeting the ROCK-AKT pathway in advanced solid tumors. There are many drawbacks of using ROCK inhibitors to target the ROCK–myosin II pathway: the activation of NM IIA via other mechanisms, the systemic toxicity of ROCK inhibitors, and the lack of isoform-specific ROCK inhibitors. Although the ROCK–myosin II pathway is important for tumorigenesis, researchers have not been able to translate these preclinical findings to clinical applications. Currently, many researchers are exploring the potential of NM IIA as a therapeutic target in tumors and other diseases.

## Figures and Tables

**Figure 1 ijms-25-09435-f001:**
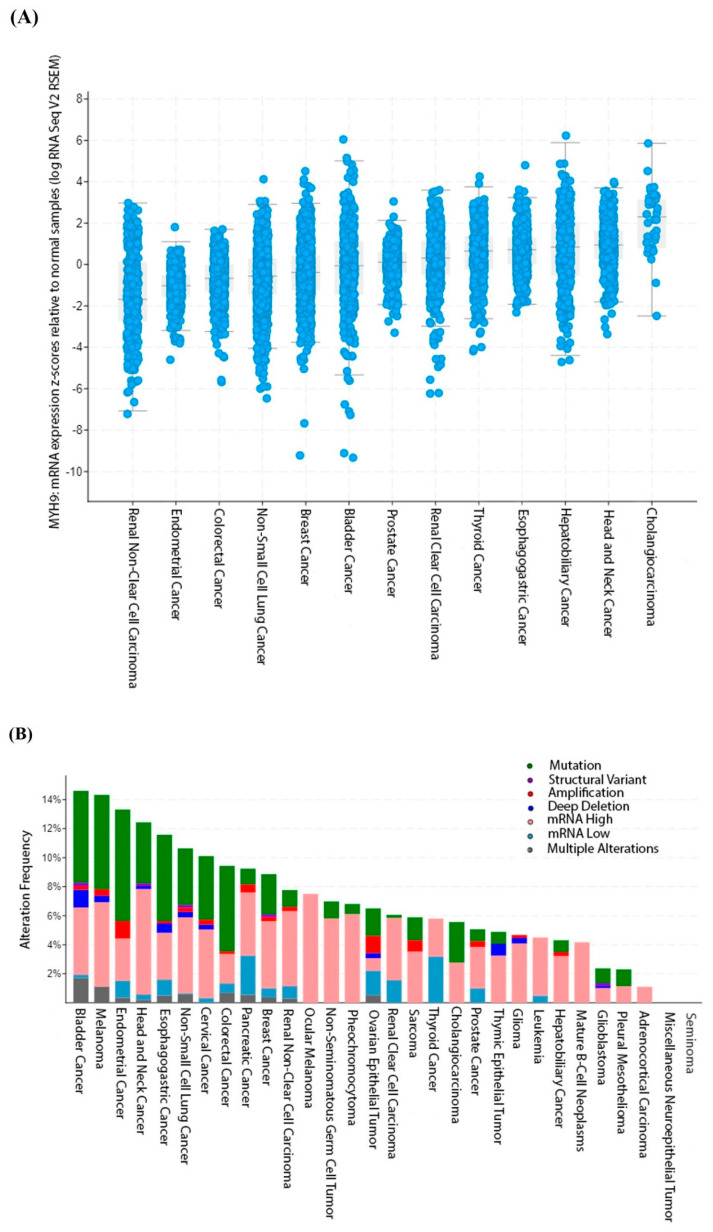
The expression and alteration profile of *MYH9* at the pancancer level. (**A**) *MYH9* mRNA expression from 32 TCGA datasets. (**B**) The alteration profile of *MYH9* from the same 32 TCGA datasets. The expression and alteration frequency data were obtained from cBioPortal (www.cbioportal.org).

**Figure 3 ijms-25-09435-f003:**
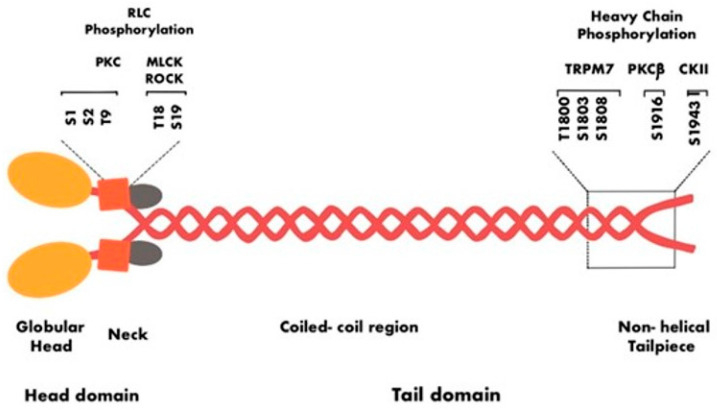
The figure shows the specific kinases involved in the phosphorylation of serine and threonine residues of both RLCs and heavy chains. PKC, protein kinase C; MLCK, myosin light chain kinase; ROCK, Rho-associated protein kinase; TRPM7, transient receptor potential melastatin 7; PKCβ, protein kinase Cβ; CK II, casein kinase II. The figure was adapted from Pecci et al. [28].

**Figure 4 ijms-25-09435-f004:**
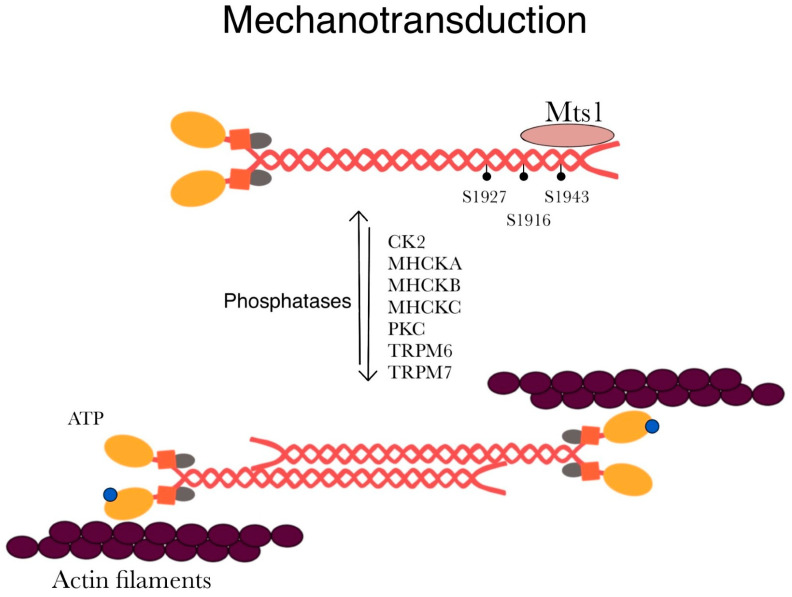
This figure illustrates the regulation of myosin II filament formation and activity. Specific serine residues on the myosin heavy chain (S1916, S1927, and S1943) are phosphorylated by various kinases, including MHCKA, MHCKB, MHCKC, TRPM6, TRPM7, PKC, and CK2. The phosphorylation of these sites is depicted as favoring the filamentous state of myosin, which is essential for mechanotransduction and ATP hydrolysis-driven interaction with actin filaments. Phosphatases are the enzymes responsible for dephosphorylation, which may reverse the phosphorylation effect, potentially leading to a shift back to the monomeric state of myosin. Mts1, also known as S100A4, is a calcium-binding protein that regulates myosin II function by modulating filament assembly. It binds to the myosin heavy chain, influencing the balance between monomeric and filamentous forms of myosin II. Mts1 typically inhibits filament formation, thereby controlling myosin’s contractile activity and its ability to interact with actin.

**Figure 5 ijms-25-09435-f005:**
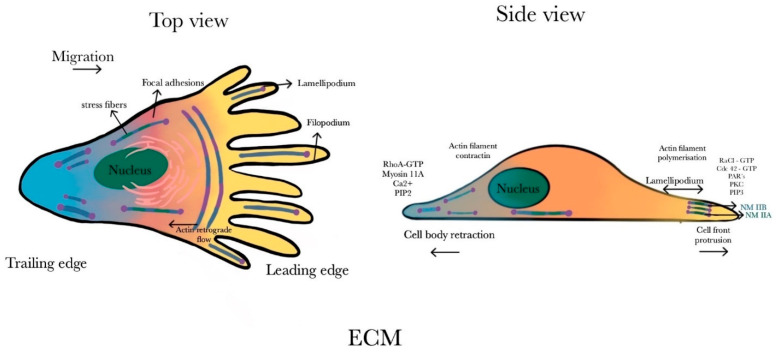
The figure shows the structure and orientation of cell migration in the 2D environment. At the front, actin filaments within lamellipodia and filopodia are oriented with their rapidly polymerizing ends in the forward direction. In the main body, actin and myosin filaments form bipolar structures to aid in cell retraction. NM IIA and NM IIB show distinct localizations inside the cell, with NM IIA predominantly being found at the leading edge where actin dynamics are most active. NM IIB is predominant toward the rear end. The region between the leading and trailing edges contains varying concentrations of NM IIA and NM IIB. Additional molecules, such as RhoA, Rac1, Cdc42, Ca^2+^ ions, and αPKC, also play significant roles in this cellular organization and migration process.

**Figure 6 ijms-25-09435-f006:**
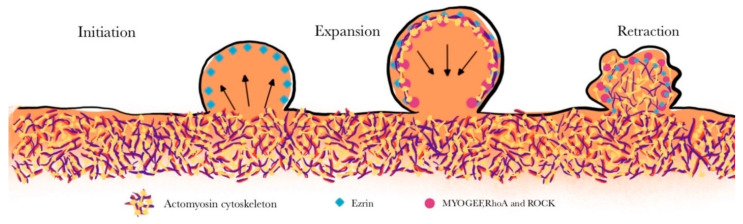
The formation of plasma membrane blebs consists of three phases: initiation, expansion, and retraction. Blebbing stimuli, such as Ca^2+^ influx and apoptosis, induce the initiation of membrane protrusion. Actomyosin contractility drives the expansion of blebs, which are devoid of the F-actin cortex. Rho-ROCK signaling then drives bleb retraction via actomyosin contractility. NM IIA contractile forces promote bleb retraction.

**Figure 7 ijms-25-09435-f007:**
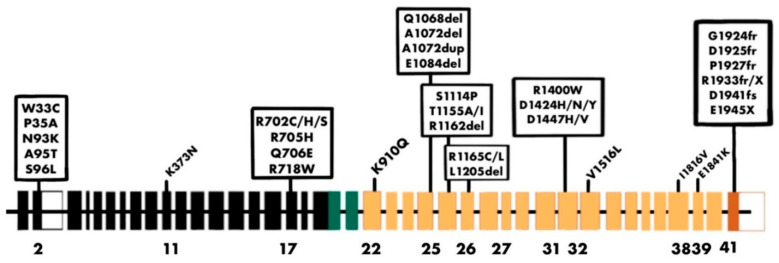
A schematic representation of the *MYH9* exons with common mutations found in patients with *MYH9*-RD. The color coding of exon organization is as follows: black, motor domain; green, neck; orange, coiled coil domain; and brown, non-helical tail.

**Table 1 ijms-25-09435-t001:** This table summarizes the key intramolecular interactions within the myosin II molecule that help maintain its 10S or “OFF” state.

Interaction	Interaction Type	Components Involved	Specific Regions	Significance
BF1	Head–Head	Blocked Head (BH) ↔ Free Head (FH)	BH: Loop I365–N381, Helix T382–L390 ↔ FH: Helix E727–Y734, Loop E735–D748	Stabilizes the myosin heads in the pre-power stroke state, preventing ATP hydrolysis and actin binding.
BF2	Head–Head	BH RLC ↔ FH RLC	N terminal lobes: BH: Helix A, A–B linker ↔ FH: Helix D, A–B linker	Strengthens head–head interaction, crucial for regulating muscle activity in the 10S state.
TB1	Head–Tail	Blocked Head (BH) ↔ Tail (Seg3)	Seg3: L1604–E1612 ↔ BH: Loop L450–F460	Provides weak electrostatic interaction that contributes to maintaining the 10S state.
TB2	Head–Tail	Blocked Head (BH) ↔ Tail (Seg2)	Seg2: L1431–D1436 ↔ BH: Helix K72–D74 (SH3 domain)	Physically blocks BH converter domain movement, preventing ATP turnover.
TB3	Head–Tail	Blocked Head (BH) ↔ Tail (Seg2)	Seg2: Q1445–L1452 ↔ BH: R718, L766 (near converter domain)	Inhibits necessary movements in the BH, trapping ATP hydrolysis products.
TB4	Head–Tail	Blocked Head (BH) ↔ Tail	Tail: L1494–L1498 ↔ BH: Helix D (ELC N-lobe)	Stabilizes 10S conformation by reinforcing the interaction between the tail and BH.
TB5	Head–Tail	Blocked Head (BH) ↔ Tail (Seg3)	Seg3: A1577–R1584 ↔ BH: Helix E (RLC)	Further stabilizes the 10S structure by anchoring Seg3 to the BH regulatory domain.
TB6	Head–Tail	Blocked Head (BH) RLC ↔ Tail (Seg3)	BH RLC N terminal extension ↔ Seg3: Residues 1560–1572	Crucial for maintaining the 10S state by strengthening the head–tail interaction, involving the phosphorylation domain (PD).
TF1	Head–Tail	Free Head (FH) ↔ Tail (Seg1)	FH: CM Loop T404–K420 ↔ Seg1: M925–A941	Inhibits FH actin binding, keeping the myosin in an inactive state.
TF2	Head–Tail	Free Head (FH) ↔ Tail (Seg1)	FH: Loop 2 K626–T658 ↔ Seg1: M925–A941	Further inhibits actin binding by the FH, reinforcing the 10S state.
TF3	Head–Tail	Free Head (FH) RLC ↔ Tail (Seg1)	FH RLC Helix A ↔ Seg1: L850–Q856	Influences regulatory movements of the FH RLC, contributing to the stability of the 10S conformation.
TT1	Tail–Tail	Segment 1 (Seg1) ↔ Segment 3 (Seg3)	Weak electrostatic interactions across several regions	Supports the compact folding of the tail, crucial for maintaining the 10S state.
TT2	Tail–Tail	Segment 1 (Seg1) ↔ Segment 3 (Seg3)	Seg1: R910–M925 ↔ Seg3: L1628–E1647	Maintains close alignment of the tail segments, reinforcing the 10S conformation.

**Table 2 ijms-25-09435-t002:** Altered NM IIA regulation in various cancer.

Tumor Type	Mechanism	References
Squamous cell carcinoma	➢ Human head and neck: dual role as tumor promoter and tumor suppressor	[200,201]
	**Tumor suppressor**:	
	➢ Mouse head and neck: tumor suppressor	[202]
	➢ Skin (mouse): tumor suppressor	[202]
	➢ Tongue (mouse): tumor suppressor	[203]
Melanoma	➢ Mouse: tumor suppressor	[204]
	**Tumor promoter**:	
Gliomas	➢ Glioma: acts as tumor promoter and interacts with THBS1 to promote glioma cell malignancy	[205]
	➢ Glioblastoma: promotes pro-tumorigenic vascular functions	[206]
Nasopharyngeal carcinoma	➢ Promotes tumor stemness and EMT, leading to NPC progression	[207,208,209,210,211]
Lung carcinoma	➢ In non-small cell lung carcinoma, NM IIA is associated with poor prognosis	[212,213]
	➢ NM IIA is involved in development of resistance to *EGFR* Tyrosine kinase inhibitors in NSCLC	[214]
	➢ NM IIA promotes malignant behavior of lung adenocarcinoma	[215]
Hepatocellular carcinoma	➢ NM IIA overexpression is associated with stemness and poor prognosis	[216]
	➢ Low NM IIA level is associated with reduced cell proliferation, metastasis, and resistance to sorafenib	[217]
Esophageal squamous cell carcinoma	➢ NM II overexpression is associated with poor prognosis	[218,219,220]
Pancreatic cancer	➢ High levels of NM IIA promote EMT and metastasis	[221]
Gastric cancer	➢ NM II overexpression is associated with poor prognosis	[222,223]
Colorectal cancer	➢ NM IIA promotes CRC metastasis by interacting with MAPK/AKT, ATG9B/ integrin β1, and LIMK/actinin-4	[224,225,226]
Breast cancer	➢ Phosphorylation of NM IIA at Ser1943 residue results in increased cell motility and chemotaxis	[95]
	➢ In TNBC cells, myosin-IIA, along with integrin β1 and MLCK, contributes to adaptive resistance development against MEK inhibitors	[227]
	➢ NM IIA modulates biophysical characteristics of ECM during cancer progression	[228]
Renal cancer	➢ NM IIA facilitates nuclear translocation of chemokine receptor CXCR4 which drives metastasis	[229]
	➢ Overexpression of NM IIA drives progression of clear cell RCC	[230]
Prostate cancer	➢ NM IIA maintains nuclear stability during cell migration	[231]
Osteosarcoma	➢ High levels of NM IIA activate Wnt/β-catenin pathway to promote tumorigenesis	[232]
Thyroid cancer	➢ In papillary thyroid carcinoma, NM IIA promotes secretion of matrix metalloproteinase 1 (MMP1) to increase cell proliferation and metastasis	[233]
	➢ In differentiated thyroid carcinoma, high levels of NM IIA contribute to development of radioresistance	[234]
Acute myeloid leukemia	➢ Increased NM IIA expression enhances actomyosin contractility of AML cells	[235]
Diffuse large B cell lymphoma	➢ NM IIA activates Wnt-β-catenin-STAT3 pathway to promote disease progression	[236]

**Table 3 ijms-25-09435-t003:** Drugs targeting *MYH9*.

Drugs	Mode of Action	References
Blebbistatin	➢ Non-competitive myosin-II inhibitor	[320]
Cinobufotalin	➢ Reduces expression of *MYH9* in HCC and lung cancer	[217,255]
Apatinib	➢ Inhibits phosphorylation of *VEGFR2* in glioma cells	[205]
Disulfiram	➢ Directly targets *MYH9* and Ras GTPase-activating-like protein IQGAP1 to inhibit growth of TNBC cells	[323]
C_70_-EDA	➢ Inhibits NM IIA filament assembly in A549 lung adenocarcinoma cells	[324]
J13	➢ Disrupts *MYH9* interaction with actin	[325]
Staurosporine	➢ Inhibits phosphorylation of *MYH9* at Ser1943 residue in gastric cancer	[222]
Homoharringtonine	➢ Upregulates *MYH9* expression in AML and CML cell lines	[326]
Fasudil	➢ Rho-kinase inhibitor and calcium channel blocker	[327]
Ripasudil and Netarsudil	➢ ROCK1 and ROCK2 inhibitor	[328,329]
Belumosudil	➢ Selective ROCK2 inhibitor	[330,331]
Y27632	➢ ROCK1 and ROCK2 inhibitor	[332]
GSK269962A	➢ Selective ROCK1 inhibitor	[333]
AT13148	➢ Dual ROCK/AKT inhibitor	[334]
BDP5290	➢ MRCK inhibitor	[335]
Rhodblock 6	➢ Rho kinase inhibitor	[336]
RKI-1447 and RKI-18	➢ Rho kinase inhibitor	[337,338]
Thiosemicarbazone iron chelators	➢ Inhibits ROCK1/p-MLC2 pathway	[339]
Combretastatin (CA-4)	➢ Increases RLC phosphorylation	[340]

## Data Availability

Data for this review were collected through PubMed. The following search terms were used: non-muscle myosin IIA; NM IIA; MYH9; tumor; cancer; MYH9-RD; myosin II; MYH9 mutations; oncogene; tumor suppressor; and actomyosin. Only articles published in English were included in the literature review. All relevant data are incorporated into the manuscript.

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
