# Peer review of "Non-Muscle Myosin II A: Friend or Foe in Cancer?"

_ijms, 2024, doi:10.3390/ijms25179435_

Round 1

Reviewer 1 Report

Comments and Suggestions for Authors

The authors review the role of non-muscle myosin IIA in cancer. The first half focusses on the structure, function and regulation of this myosin while the latter part discusses the role of NMIIA in specific cancers in a detailed way. The review appears to be quite comprehensive and will be valuable to the scientific community for its breadth of topics. I have comments on myosin structure that should be incorporated into a revised version for accuracy.

Specific comments;

Line 46 “Myosin superficially binds to actin filaments…”. Superficially is a poor choice of words and should be deleted.

Line 86 Delete “In contrast”, which is not appropriate.

Line 94 The UCS chaperone is necessary to fold the heavy chain but not the light chains.

Line 121 The light chains bind in a calcium independent manner but not calmodulin.

Line 139 The proline kink allows folding but does not cause it.

Line 144 The 10S form exhibits essentially NO actin-activated ATPase activity, so this sentence is incorrect.

Line 151 “…the RLC is important for ATP hydrolysis via the myosin II motor complex”. The RLC is important to maintain non-aggregated myosin that can form filaments but does not directly contribute to ATP hydrolysis.

Line 205 “Rho kinase can reduce the activity of protein phosphatase I by phosphorylating the RLC to increase the NMII activity.” I am not sure what the authors mean to say. Do they mean that decreased phosphatase activity leads to enhanced RLC phosphorylation?

Fig 2 Although this is a schematic, the cartoon of the 10S conformation is very misleading although I have seen similar depictions in other publications. There are interactions between the segments of the rod that are not indicated by the circle of the tail depicted here. There are recent cryo-EM structures of the inhibited state that can serve as guides to present a more accurate schematic of the 10S state (M Peckham lab, R Craig lab, and SM Heissler) that show that rod interactions occur that stabilize the off state.

Fig 4 This figure is not very clear to me. Mts1 is not defined in the legend. The word “phosphatases” is sort of dangling off the left side of the figure. Whether phosphorylation favors monomers or filaments is not clear.

On the cancer side of the review, I don’t see any references to some interesting studies by SS Rosenfeld and colleagues on the role of non-muscle myosin II A and B in glioblastomas. The authors should add these.

Reviewer 2 Report

Comments and Suggestions for Authors

The manuscript „Non-muscle myosin II A: Friend or Foe in Cancer?”, written by Feroz W, Park BS, Siripurapu M, Ntim N, Kilroy MK, Sheikh AA, Mishra R and Garrett JT. is a review describing the structure and the roles of non-muscle myosin II A (NM IIA) in normal tissue and different types of tumors. The manuscript begins with the short overview of the topic and presents the structure of the NM II A, its regulation by different signaling pathways, its physiological functions, roles of its heavy chain gene, MYH9, as an oncogene in different types of tumors, roles of MYH9 as a tumor suppressor, roles of NM IIA in other pathological conditions, description of drugs targeting MYH9, directly or indirectly and perspectives in future investigations.

The manuscript is well organized and well written, informative, with detailed descriptions of the myosin functions and structure, illustrated with schematic figures and tables which have appropriate legends. There are more than 300 references.

The signaling pathways are described in details. However, it would be interesting to describe in more details how can MYH9 increase activity of MAP kinases and wnt signaling (and how can MYH9 ubiquitinate GSK3 beta). Also, in that part of the manuscript, there is description of the interactions between a gene (MYH9, as genes are written in italics) and proteins.

Minor corrections: lines 10, 549, 566, 788, 790, 822, 955.
